# MouseGoggles: an immersive virtual reality headset for mouse neuroscience and behavior

Matthew Isaacson ®[1,3] ✉, Hongyu Chang ®[2,3], Laura Berkowitz[1], Rick Zirkel ®[1], Yusol Park[2], Danyu Hu[1], Ian Ellwood[2,4] & Chris B. Schaffer ®[1,4]

Small-animal virtual reality (VR) systems have become invaluable tools in neuroscience for studying complex behavior during head-fixed neural recording, but they lag behind commercial human VR systems in terms of miniaturization, immersivity and advanced features such as eye tracking. Here we present MouseGoggles, a miniature VR headset for head-fixed mice that delivers independent, binocular visual stimulation over a wide field of view while enabling eye tracking and pupillometry in VR. Neural recordings in the visual cortex validate the quality of image presentation, while hippocampal recordings, associative reward learning and innate fear responses to virtual looming stimuli demonstrate an immersive VR experience. Our open-source system's simplicity and compact size will enable the broader adoption of VR methods in neuroscience.

Virtual reality (VR) systems for laboratory animals have enabled fundamental neuroscience research, supporting the study of neural processes underlying complex cognitive tasks using neural recording strategies that require head fixation[1–4]. VR gives the experimenter full control over the subject's visual experience and allows experimental manipulations infeasible with real-world experiments, including teleportation and visuomotor mismatch paradigms[4]. VR with head-fixed mice has traditionally relied on panoramic displays composed of projector screens[1,3] or arrays of light-emitting diode (LED) displays[2,4] positioned 10–30 cm away from the eyes to remain within the mouse's depth of field. This necessitates displays that are orders of magnitude larger than the mouse, resulting in complex, costly and light-polluting systems that can be challenging to integrate into many neural recording setups. In addition, fixed experimental equipment (for example, cameras, lick ports and microscope objectives) can obstruct the mouse's visual field, potentially reducing immersion in the virtual environment. Inspired by modern VR solutions for humans, we set out to design a headset-based VR system for mice to overcome the constraints of panoramic VR.

## Results

### Miniature VR headset design

Using small circular displays and short-focal length Fresnel lenses, we designed eyepieces suited to mouse eye physiology (Fig. 1a). Spherical distortion of the display by the lens results in a near-constant angular resolution of 1.57 pixels per degree and Nyquist frequency of 0.78 cycles per degree (c.p.d.)—just above the 0.5 c.p.d. spatial acuity of mouse vision[5]—and a field of view (FOV) coverage spanning up to 140° (Fig. 1b,c) per mouse eye. The optical design positions the display near infinity focus (Fig. 1d), estimated as the optimal focal length for mouse vision, making the clarity of image presentation robust to small deviations in eye position, as validated by imaging gratings projected on the back of an enucleated mouse eye (Extended Data Fig. 1). Using two eyepieces separated to accommodate a typical mouse intereye distance (Extended Data Fig. 2), we achieve 230° horizontal FOV coverage with ~25° of binocular overlap, and 140° vertical FOV coverage spanning −55° to 85° elevation at a headset pitch of 15°. This configuration covers a large fraction of the mouse's visual field, which we approximate from a previous study[6] to span 180° in azimuth and 140° in elevation

[1]Meinig School of Biomedical Engineering, Cornell University, Ithaca, NY, USA. [2]Department of Neurobiology and Behavior, Cornell University, Ithaca, NY, USA. [3]These authors contributed equally: Matthew Isaacson, Hongyu Chang. [4]These authors jointly supervised this work: Ian Ellwood, Chris B. Schaffer. ✉e-mail: mdi22@cornell.edu

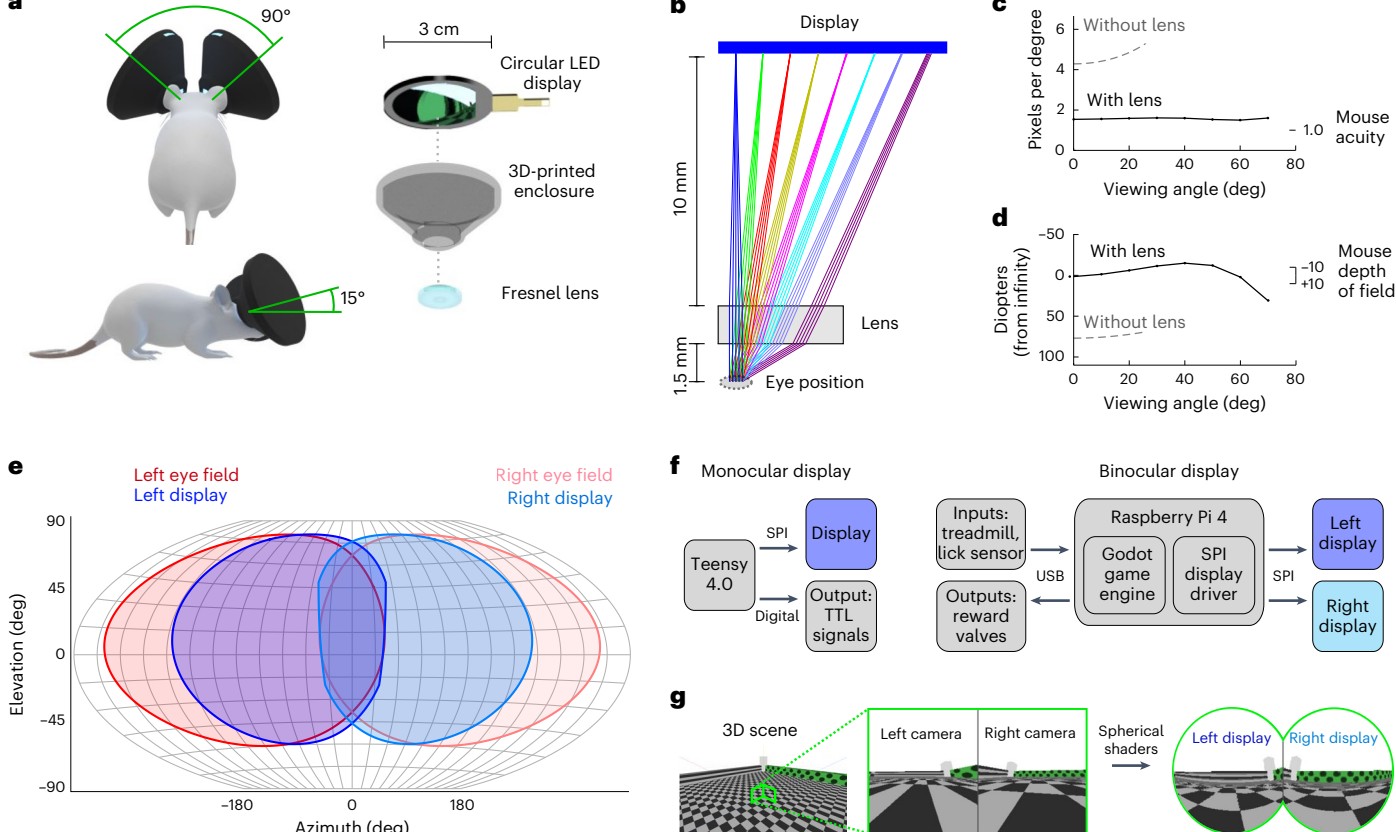

**Fig. 1 | Headset-based VR design. a**, Components and orientation of headset eyepieces, each containing a 2.76-cm-diameter circular LED display and 1.27-cm-diameter Fresnel lens housed in a 3D-printed enclosure. **b**, Optical modeling of display and Fresnel lens for infinity focus, with viewing angles of 0–70° (one half of the 140° total FOV coverage) linearly mapped onto the circular display. **c,d**, Optical model estimate for the apparent resolution (**c**) and focal distance (**d**) as a function of viewing angle. **e**, A Winkel tripel projection of the mouse's estimated visual field overlaid with the headset display visual field coverage. **f**, Communication diagrams of the MouseGoggles Mono monocular display system (left) and MouseGoggles Duo binocular display system (right), with SPI-based display control and additional input/output communication schemes. **g**, The Godot video game engine-generated 3D environment with split-screen viewports and spherical shaders to map the scene onto the dual-display headset.

per eye, with the optical axis centered on 70° azimuth and 10° elevation (Fig. 1e). Unique features of this headset-based system are the independent control over each eye's display (allowing stereo correction[7]) and the ability to adjust headset pitch to enable greater overhead stimulation—an understudied area of vision in head-fixed VR contexts probably important for prey animals.

To generate images and video for the eyepiece displays, we designed two types of control systems. The first connects a single display to a high-speed microcontroller, ideal for simple monocular visual stimulation experiments commonly used in vision neuroscience (Fig. 1f). The second connects two displays to a Raspberry Pi 4 using a split-screen display driver (Fig. 1f). We used the user-friendly video game engine Godot to quickly build three-dimensional (3D) environments, program experimental paradigms and perform low-latency input/output communication to external equipment with frame-by-frame synchronization. With a two-eye viewport and custom shaders to map the 3D environments onto the eyepieces (Fig. 1g), MouseGoggles generates high-performance VR scenes at 80 fps and <130 ms input-to-display latency during full-screen updates. The entire monocular and binocular display systems can be housed in a single enclosure of 3D-printed parts or in smaller headset form factors by separating the Raspberry Pi from the eyepieces (Extended Data Fig. 3 and Supplementary Video 1). This compact design enables more mobile and rotatable VR systems (Supplementary Video 2) but results in partial occlusion of the mouse's whiskers, with slightly more or less occlusion depending on the headset pitch angle (Extended Data Fig. 4).

## Validation of image presentation

To validate the function of our eyepiece design, we delivered visual stimulation with the monocular display, named MouseGoggles Mono, to anesthetized, head-fixed mice during two-photon calcium imaging of the visual cortex (Fig. 2a). When using blue stimuli to excite V1 neurons, thereby reducing spectral overlap with green GCaMP6s fluorescence[8], we found that the monocular display produced 99.3% less stray light contamination into the fluorescence imaging channels than a traditional unshielded LED monitor (Fig. 2b). Total stray light from the eyepiece display was equivalent to that produced by a carefully shielded monitor, and we did not detect any additional light reaching the light detection systems by entering the pupil and scattering through the brain (Extended Data Fig. 5).

Presenting drifting bars and gratings elicited orientation- and direction-selective responses (Fig. 2c) from which we calculated stimulus tuning properties of primary visual cortex layer 2/3 (V1 L2/3) neurons nearly identical to those previously obtained with traditional displays, such as a median receptive field (RF) radius of 6.2° (versus 5–7° with a monitor[9]) (Fig. 2d,e), maximal neural response at a spatial frequency (SF) of 0.042 c.p.d. (versus 0.04 c.p.d. (ref. 10)) (Fig. 2f,g), and a median semisaturation contrast of 31.2% (versus 34% (ref. 11)) (Fig. 2h,i), demonstrating that the display produces in focus, high-contrast images for the mouse visual system.

To validate the efficacy of the binocular MouseGoggles system, named MouseGoggles Duo, for simulating virtual environments, we displayed a linear track to awake, head-fixed mice positioned on a

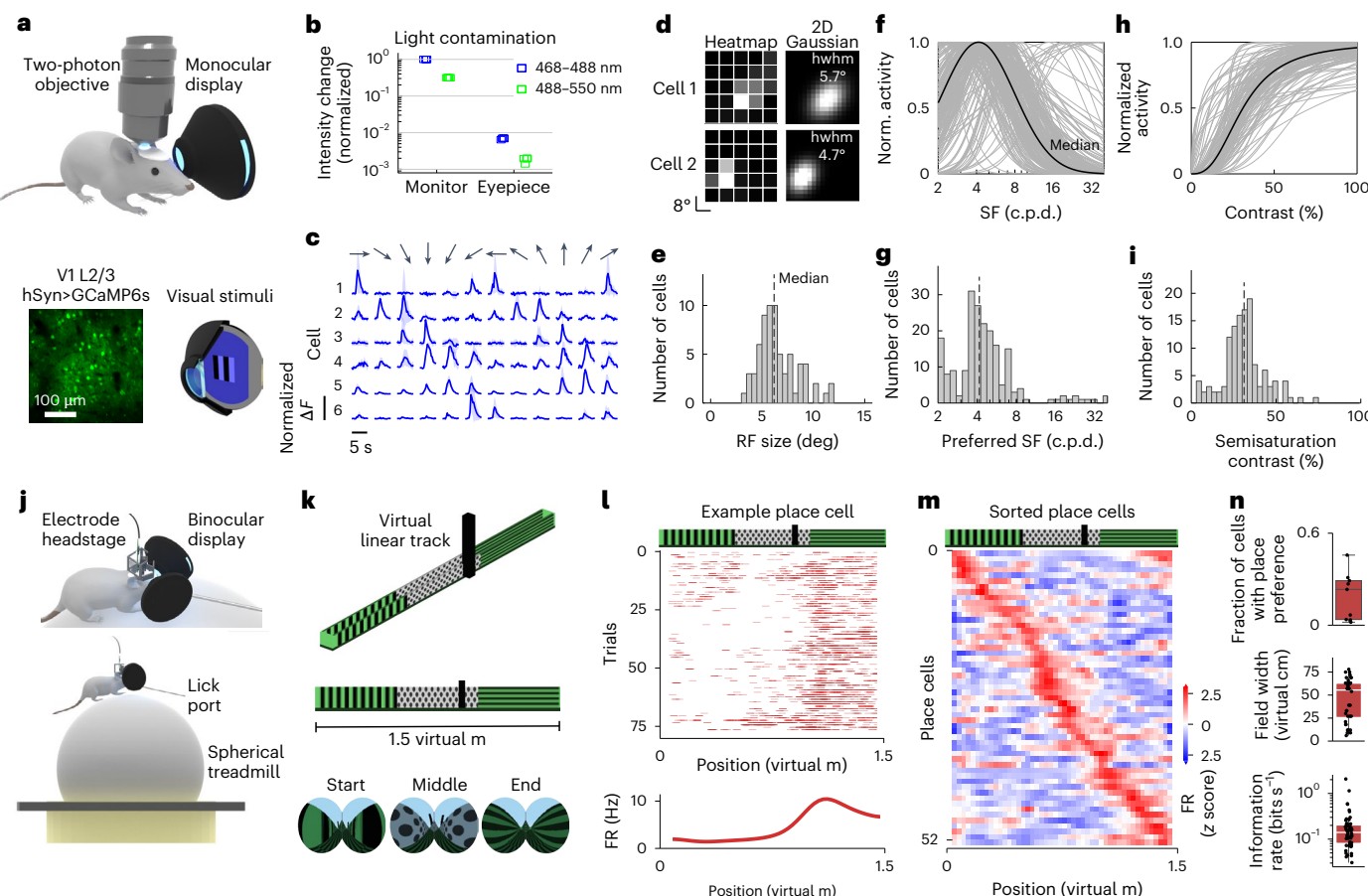

**Fig. 2 | Neural recording in headset VR. a**, The experimental setup for MouseGoggles Mono visual stimulation with two-photon imaging of mouse V1 layer 2/3 neurons expressing GCaMP6s. **b**, Light contamination measurements from five repetitions of a maximum-brightness blue flicker stimulus into blue (468–488 nm) and green (488–550 nm) imaging channels, using either a flat LED monitor or the monocular display eyepiece. Raw intensity values were normalized to the maximum intensity from the monitor. **c**, Direction- and orientation-selective fluorescence change (ΔF) responses from 6 example neurons from 12 directions of drifting grating stimuli (mean ± s.d. of 6 repetitions). **d**, RF maps for two example cells. Left: inferred spike rate heatmap based on stimulus location. Right: 2D Gaussian fit to the heatmap, with the average half width at half maximum (hwhm) shown. **e**, A histogram of calculated RF size for all cells well fit by a 2D Gaussian (n = 341 cells). **f**, The SF tuning of normalized activity for all cells well fit by a log-Gaussian function (n = 124 cells). **g**, A histogram of preferred SF. **h**, The contrast frequency tuning of normalized activity for all cells well fit by a Naka–Rushton function (n = 202). **i**, A histogram of semisaturation contrast. **j**, The experimental setup for hippocampal electrophysiological recordings during simulated walking on a spherical treadmill with MouseGoggles Duo. **k**, Rendered view (top) and side view (middle) of the virtual linear track, with headset views at three different positions (bottom). **l**, An example place cell across the entire virtual linear track session, showing the raster plot of neural activity (top) and tuning curve (bottom; FR, firing rate). **m**, A position-ordered heatmap of all detected place cells (n = 54 cells), showing binned firing rate (FR, z-scored) over position. **n**, Place cell characteristics over all recorded sessions: fraction of cells with place selectivity (n = 9 sessions, top), place field width (n = 39 cells within 10–80 virtual cm, middle) and information rate (n = 54 cells, bottom). The box plot displays median and 25th and 75th quartiles, with the whiskers representing the most extreme nonoutliers.

spherical treadmill while simultaneously performing electrophysiological recording of hippocampal cornu ammonis (CA1) neurons (Fig. 2j,k). Place fields developed over the course of a single session of virtual continuous-loop linear track traversal (Fig. 2l), with place cells (19% of all cells versus 15–20% with projector VR[12]) found to tile the entire virtual track over multiple recording sessions (Fig. 2m). Many place cells with high spatial information encoded field widths as small as 10–40 virtual cm, or 7–27% of the total track length, but we also observed larger field widths of 50–80 virtual cm, which more closely match the size of visually distinct zones of the track (Fig. 2n). Taken together, these data demonstrate that MouseGoggles effectively conveys virtual visual and spatial information to head-fixed mice.

### Learned and innate behaviors in immersive VR

To assess our ability to condition mouse behaviors in headset VR, we trained mice on a 5-day continuous-loop linear track place learning protocol in which mice were given liquid rewards for licking at a specific

virtual location using MouseGoggles Duo (Fig. 3a and Extended Data Fig. 6). After 4–5 days of training in the linear track, mice exhibited increased anticipatory licking (licking inside the reward zone just before a reward) and reduced exploratory licking in an unrewarded control zone (Fig. 3b). We trained two mouse cohorts with different reward locations in the same virtual linear track and found a statistically significant increase in lick preference in the reward zones during unrewarded day 4–5 probe trials (two-tailed Mann–Whitney U test, P = 0.02; Fig. 3c,d), demonstrating spatial learning in VR similar to previous results with a projector-based system[13].

A potential benefit of MouseGoggles over panoramic displays is a greater degree of immersion in the virtual environment, as the headset effectively blocks irrelevant and conflicting visual stimuli. To determine whether innate behavioral responses can be elicited by more immersive head-fixed VR, we presented looming visual stimuli to naive mice that had no prior experience with head-fixed displays (Fig. 3e). On the first presentation of a looming stimulus

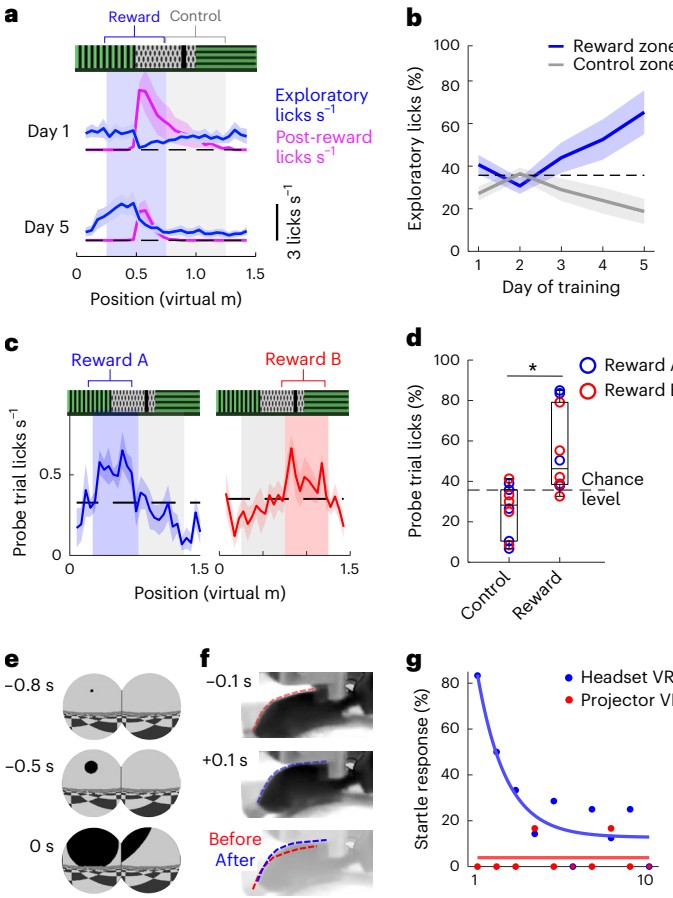

**Fig. 3 | Conditioned and innate behaviors in headset VR. a,** Mouse licking behavior during days 1 and 5 of a 5-day virtual linear track place learning protocol using MouseGoggles Duo, separated into an exploratory licking (defined as licks not initiated by a liquid reward delivery) and post-reward licking (mean of all trials ± s.e.m., $n = 5$ mice). **b,** The proportion of exploratory licks in reward versus control zone, across days (mean of all trials ± s.e.m., $n = 5$ mice). **c,** The exploratory lick rate during unrewarded probe trials on days 4 and 5 as a function of position in the virtual linear track for mice trained for a reward in zone A (left) and zone B (right) (mean lick rate of probe trials ± s.e.m., $n = 5$ mice each for reward zone). **d,** The proportion of licking in reward versus control zone on probe trials in which no reward was delivered, pooling mice conditioned to associate reward with zones A and B. The box plots display median and 25th and 75th quartiles, with the whiskers representing the most extreme nonoutliers ($n = 10$ mice; median 46.3% for reward versus 28.6% for control; $P = 0.02$, two-tailed Mann–Whitney $U$ test). **e,** Looming stimuli consisting of a dark circular object approaching at constant velocity before reaching the closest distance at $t = 0$ s. **f,** An example 'startle' response from a head-fixed mouse from the looming stimulus, characterized by a jump up and arching of the back (see also Supplementary Video 3). **g,** The proportion of mice that displayed a startle response after presentation of a looming stimulus (as determined from manual behavior scoring) as a function of stimulus repetition and comparing headset or projector-based VR ($n = 6$–7 mice for all repetitions with headset, $n = 4$–5 mice with projector). An exponential decay curve is fit to the headset-based VR startle responses.

using MouseGoggles Duo, nearly all mice displayed a head-fixed startle response (a rapid jump or kick, with an arched back and tucked tail, manually scored; Fig. 3f and Supplementary Video 3), while a nearly identical experiment on a traditional projector-based VR system (Extended Data Fig. 7) produced no immediate startles. Startle responses were found to rapidly extinguish with repeated looming stimuli (Fig. 3g), an adaptation previously observed with defensive responses to looming in freely walking mice[14].

## VR with integrated eye and pupil tracking

To enable monitoring of eye and pupil dynamics while mice experience the VR environment, we developed a binocular headset, named MouseGoggles EyeTrack, with infrared (IR)-sensitive cameras embedded in the eyepieces (Fig. 4a). We used this headset with head-fixed mice walking on a small-footprint linear treadmill[15] (Fig. 4b). Each eye is seen through reflection on an angled hot mirror and spectrally separated from the display by software-based removal of red light from the VR scene (Fig. 4c). We calibrated this system to account for visual distortion from the Fresnel lens and tilted camera view (Fig. 4d), and used Deeplabcut[16] for offline tracking of points on the eyes and pupils to enable absolute measurements of pupil diameter and position (Fig. 4e). Assuming typical values for eye distance from the lens and eye size, we estimated the eye's optical axis relative to the virtual visual field (Fig. 4f). Presenting looming stimuli with MouseGoggles EyeTrack (Fig. 4f and Supplementary Video 4) led to a sharp slowdown or reversal of forward walking (Fig. 4g,h) and vertical shifts in gaze position during the overhead loom (Fig. 4i,j). Since eye movements during head fixation are known to be associated with attempted head movements[16], these gaze shifts following the looming stimulus may in part be related to attempted escape behaviors. Pupil diameter was also found to increase after the loom (Fig. 4k,l), but unlike the walking slowdown and gaze-shift behaviors that persisted across repeated loom stimuli, pupil dilation responses were found to diminish with additional repetitions of the stimulus (Cuzick's trend test, $P = 0.007$), similar to the behavioral adaptation of the startle response (Fig. 3g).

## Discussion

Our headset-based VR system demonstrates substantial improvements over panoramic display systems, enabling traditional mouse neuroscience and behavioral experiments as well as opening the door to more experiments of innate behaviors during head fixation. Since the original submission of this study, two other headset-based VR systems have been described, Moculus[17,18] and iMRSIV[19]. Collectively, these systems reinforce the benefits of headset VR, including reduced cost and form factor, the ability to present stereoscopic 3D VR scenes and greater immersivity of the VR environment for mice, demonstrated by rapid visual learning in a pattern discrimination task and cliff avoidance in a virtual elevated maze test with Moculus[17], and fear responses to looming visual stimuli with iMRSIV[19] and MouseGoggles.

Although the benefits of headset-based VR are substantial, our MouseGoggles system has some limitations. Partial occlusion of the whiskers (depending on headset pitch) may confound the sensory experience of virtual navigation. Input-to-display latency may be too slow to support closed-loop experiments with fast behaviors (for example, eye movements) or neural events, although latency can be substantially reduced with future display driver optimizations. The simple optical design of MouseGoggles eyepieces, while making clear image presentation robust to precise eye position, limits the FOV coverage to 140°, whereas the more complex optical designs of Moculus and iMRSIV can support increased FOV (although at the expense of requiring more precise alignment of the display to the eye)[17,19]. Finally, the low-resolution displays used here, while suited for mice, may be insufficient for animals with increased visual acuity (for example, rats or tree shrews) and would need to be replaced with higher-resolution displays. By contrast, the simple low-cost design of MouseGoggles enables high-performance mouse VR using a single Raspberry Pi computer, with no external graphics processing unit required to run the feature-rich Godot 3D game engine. This enables greater scalability and increased throughput for VR training and experiments, with a complete VR setup (using a linear treadmill) fitting within a 14 × 14 cm footprint. Finally, our integration of IR cameras facilitates video-oculography-based eye and pupil tracking during VR. MouseGoggles establishes a flexible platform to further improve and expand mouse VR technologies, such

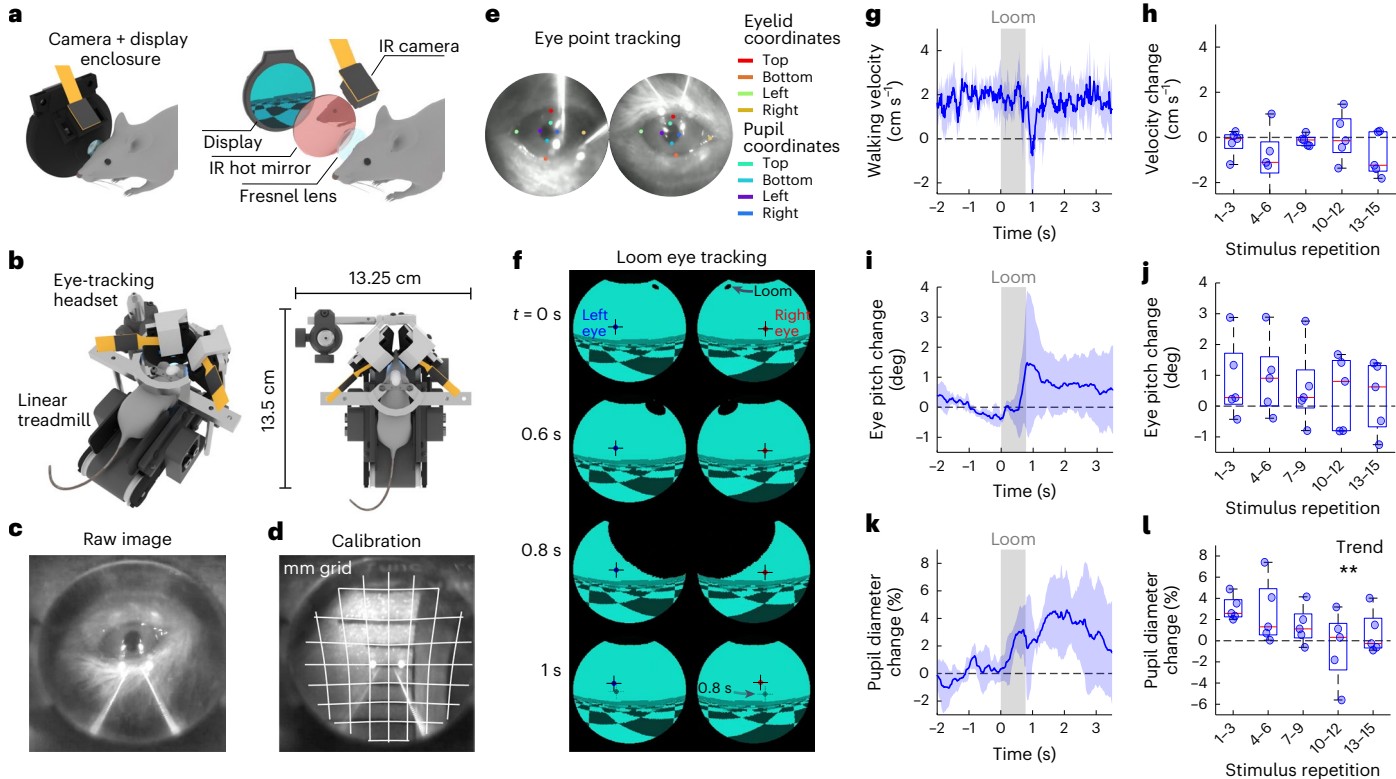

**Fig. 4 | Eye and pupil tracking during VR looming stimuli. a**, The design of combined VR display and eye-tracking camera eyepiece enclosure (left), with an exploded view showing the layout of eyepiece components (right). **b**, The experimental setup of MouseGoggles EyeTrack for head-fixed mice walking on a linear treadmill (treadmill model adapted from ref. 15). **c**, A raw IR image of the eye-tracking camera during VR. **d**, An eye-tracking camera image of a grid with calibrated gridlines (millimeter spacing), to correct distortions. **e**, Eye-tracking camera frames of left and right eyes with Deeplabcut-labeled points on the border of the eyelid and pupil. **f**, Example frames of a centered looming stimulus, with left and right eye optical axes mapped onto the VR visual field, demonstrating increased eye pitch after the loom (bottom). **g**, The average treadmill velocity during looming stimuli onset across all 15 repetitions of the stimulus (mean ± s.d. of 5 mice in shaded region). **h**, A box plot of mean walking velocity change during 0–2 s after looming stimulus onset (relative to baseline), averaged across each set of three looming stimuli (left, right and centered loom),

with no significant trend over repeat repetitions (*n* = 5 mice, one-sided Cuzick's trend test, *P* = 0.37). The box plot displays median and 25th and 75th quartiles, with whiskers representing the most extreme nonoutliers. **i**, The average change in eye pitch angle (relative to baseline) during looming stimuli across all 15 repetitions of the stimulus (mean ± s.d. of 5 mice in shaded region). **j**, A box plot of the change in eye pitch angle during 0–2 s after looming stimulus onset, averaged across each set of three looming stimuli, with no significant trend over repeat repetitions (*n* = 5 mice, one-sided Cuzick's trend test, *P* = 0.24). The box and whiskers are defined as in **h**. **k**, The average change in pupil diameter during looming stimuli onset across the first set of three repetitions of the looming stimulus (mean ± s.d. of five mice in shaded region). **l**, A box plot of the average change in pupil diameter during 0–3 s after looming stimulus onset, averaged across each set of three looming stimuli, with stars denoting a statistically significant trend (*n* = 5 mice, one-sided Cuzick's trend test, *P* = 0.007). The box and whiskers are defined as in **h**.

as in multisensory VR applications (for example, with whisker stimulation[20]), in rotatable VR setups[21] to engage or manipulate the vestibular system during VR or even in free-walking VR with further miniaturization of the headset. With our MouseGoggles system, we prioritized a design that is easy for new users to assemble and install to facilitate replication, modification and future development.

## Online content

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

## Methods

### Optical design

Optical modeling of the VR eyepiece was performed using Optic-Studio in sequential mode, with custom scripts written with Matlab (version 2022b) used for analysis and plotting. The Fresnel lens model was supplied by the manufacturer (FRP0510, Thorlabs). The display was positioned at the focal length of the lens (10 mm) for infinity focus. We estimate this to be near the center of a mouse's depth of field on the basis of previous research testing the impact of various focal length lenses on free-walking mice in a rotational optomotor assay, where it was found that either no lens or a +7 D lens with the display at a distance of ~30 (20–40) cm resulted in the strongest behavioral reactions, whereas lenses outside of this range negatively affected optomotor responses[22]. These data suggest that infinity focus (equivalent to a +3.33 D lens with the display at 30 cm) is near the center of the mouse's depth of field. Using the optical model set to infinity focus, the apparent display resolution and focal distance was estimated by first casting parallel rays from the eye position (that is, rays that appear at infinity focal depth) for multiple viewing angles (0–70°, in 10° increments). For each viewing angle, the mean position of the rays as they intersect with the display (in pixels from the display center) was calculated, and the true focal point of the rays was calculated from the position with minimum variance in ray spread. The resolution by viewing angle was calculated from the slope of the line of viewing angle as a function of pixel position. The focal depth by viewing angle was calculated from the distance between each viewing angle's focal point and the display; the inverse of this distance (in cm) quantifies the focal distance in diopters away from infinity focus. Two-dimensional (2D) projections of the visual field coverage of the display, as seen through the lens, were estimated assuming a constant display resolution of 1.57 pixels per degree: pixels were mapped onto a sphere with the center of the display pointing straight ahead, then rotated to match the final position in a typical headset orientation (45° azimuth, 15° elevation). The mouse's FOV (shown in Fig. 1 and Extended Data Figs. 4 and 7) was approximated on the basis of prior measurements of V1 retinotopic organization[6] that found RF centers roughly spanning from 0° to 140° in azimuth and −40° to 60° in elevation. Extending these RF centers by a radius of 20°, we approximate the mouse FOV as a 180° × 140° ellipse centered at 70° azimuth and 10° elevation (probably overestimating the FOV in the lower periphery where RF centers were not found).

### Display hardware

For the MouseGoggles Mono display used for V1 imaging, a circular, 16-bit color TFT LED display (TT108RGN10A, Shenzhen Toppop Electronic) was connected to a Teensy 4.0 microcontroller (Teensy40, PJRC) using a custom printed circuit board, with a short focal length Fresnel lens (FRP0510, Thorlabs). For the MouseGoggles Duo headset, two of the same displays were connected to a Raspberry Pi 4 (Pi 4B-2GB, Adafruit); both displays were connected to the serial peripheral interface (SPI) 0 port with different chip select pins (display 0 on CE0, display 1 on CE1) to allow independent display control. For the MouseGoggles EyeTrack headset, a slightly larger circular display was used (1.28 inch liquid crystal display module, model 19192, Waveshare). Plastic enclosures used to house the components (Teensy/Raspberry Pi, printed circuit board, displays, Fresnel lenses) were printed using a 4K resin 3D printer (Photon Mono X, Anycubic).

### Display software

To present visual stimuli on MouseGoggles Mono, a custom Arduino script was written for the Teensy 4.0 microcontroller and uploaded via the Arduino IDE (version 1.8.15) using the Teensyduino add-on (version 1.57). Pattern drawing commands are read over serial communication from a host personal computer (PC) and utilize the Adafruit graphics functions library (https://github.com/adafruit/Adafruit-GFX-Library) to create simple visual stimuli such as drifting gratings, edges and

flickers. To control the display from a host PC, custom scripts written with Matlab and Python (version 2.8.8) were used. To render 3D scenes for MouseGoggles Duo, the Linux-compatible game engine Godot (version 3.2.3.stable.flathub) was installed on the Raspberry Pi OS (based on 32-bit Debian Bullseye) using the Flathub Linux-based app distribution system. Unlike the Unity game engine commonly used for neuroscience VR applications, Godot was selected for MouseGoggles because it is an open-source engine with reduced computational requirements that can operate at high framerates on simple hardware (for example, a Raspberry Pi 4), despite its feature-rich and powerful 3D rendering capabilities. The 3D environment was mapped onto the circular displays using custom Godot shaders to warp the default rendered view (which linearly maps a flat plane in the virtual scene onto the flat plane of the display) to the spherical view created by the headset (which linearly maps viewing angles onto the flat plane of the display). To deliver these rendered views to the circular displays, a custom display driver was modified from an existing open-source Raspberry Pi display driver for SPI-based displays (https://github.com/juj/fbcp-ili9341), which functions by copying a subset of the default high-definition multimedia interface (HDMI) framebuffer (the size of the frame subset determined by the resolution of the SPI display) and streams the frame data over the SPI channel. Our customized driver modifies the original to enable control of two SPI displays simultaneously; this is achieved by copying a subset of the framebuffer that is twice the height of a single SPI display, with the top half defining the frame for display 0 and the bottom half for display 1. For every program loop, where the loop frequency is determined by the desired framerate, the program streams the top half of the framebuffer with chip select 0 (connected to display 0) enabled, followed by streaming the bottom half of the framebuffer with chip select 1 (connected to display 1) enabled. During each refresh cycle, the entire frame of each display is updated. Latency could be reduced in the future using the 'adaptive display stream updates' mode of the display driver (not yet implemented with MouseGoggles), where only pixels that changed from the previous frame are streamed to the display.

### Acute whole retina imaging during visual stimulation

Surgical procedure for imaging the display's projection onto the back of the enucleated mouse eye was modified from a previous method[23]. Immediately after a previously scheduled euthanasia using $CO_2$ and secondary euthanasia by cervical dislocation, curved jeweler's forceps were used to enucleate the eyeball and remove the optic nerve and connective tissue. The dissected eyeball was immediately moved into room-temperature phosphate-buffered saline. Under a stereomicroscope, the sclera, choroid and retinal pigment epithelium were removed from the eye, and the intact retina was verified visually. The eye was then placed on a 3D-printed holder with a central hole matching the typical eye diameter, with the eye's optical axis facing upward. A mini camera (OV5647, Arducam) was mounted below the eye facing upward and manually focused onto the back of the eye. Either a traditional flat monitor was mounted above the eye at a 10 cm distance, or a MouseGoggles Mono eyepiece was mounted on an adjustable optical post, with the display facing downward toward the eye. Vertically and horizontally drifting grating stimuli were then presented to the eye during by either a monitor (at a 10 cm distance) or MouseGoggles Mono display during the acquisition of retinal-plane images.

### Eye-tracking hardware and software

To perform eye and pupil tracking with the MouseGoggles EyeTrack headset, each eyepiece included additional slots for a mini IR camera module (OV5647, Arducam), an IR hot mirror (FM01, Thorlabs) placed on a 15° angle between the Fresnel lens and display, and a custom circuit board with two surface-mount IR LEDs (VSMB2943GX01, Vishay) positioned on either side of the camera module. Each camera was positioned along the side of the angled eyepiece enclosure, facing

the display, with an IR view of the mouse eye on the opposite side of the Fresnel lens based on reflection off of the hot mirror. Each camera was independently controlled using a Raspberry Pi 3 through the libcamera library (libcamera.org), acquiring 30 frames s$^{-1}$ at 800 × 800 pixel resolution. Videos were preprocessed to extract the red imaging channel (which excludes most of the blue/green VR display) of a 500 × 500 pixel region of interest centered on the eye. Offline eye and pupil tracking of this preprocessed video was performed using the Deeplabcut[16] toolbox for tracking the pixel coordinates of the top, bottom, left and right points of the pupil and eyelids. The pixel coordinates of the pupil center were calculated by averaging the left- and right-side points of the pupil, and the pupil diameter was calculated by the distance between these two points; the top and bottom points of the pupil were ignored due to their often-unreliable tracking as the mouse partially closes its eyelids. Lateral eye movements due to face movement behaviors were corrected by subtracting the movements of the eye center (defined as the average between the eyelid's left and right side point coordinates) from the pupil center, similar to previous methods[24]. Lateral movements of the pupil center in pixels were then converted to movements in millimeters using a calibrated transformation estimating the radial distortion produced by the Fresnel lens and the camera's tilted point of view. The following equations were used to relate positions (in millimeters) in the eye position plane to coordinates (in pixels) in the acquired images:

$$e'_x = e_x(a(e_x^2 + e_y^2) + b\sqrt{(e_x^2 + e_y^2)})$$

$$e'_y = e_y(a(e_x^2 + e_y^2) + b\sqrt{(e_x^2 + e_y^2)})$$

$$p_x = \arcsin\left(\frac{e'_x}{\sqrt{e_x^2 + (e_y - c)^2 + d}}\right)$$

$$p_y = \arcsin\left(\frac{e'_y}{\sqrt{e_x^2 + (e_y - c)^2 + d}}\right),$$

where $e_x$ and $e_y$ are the Cartesian coordinates of the eye plane, $a$ and $b$ are fitting parameters accounting for the radial lens distortion, $e'_x$ and $e'_y$ are the Cartesian coordinates after lens distortion, $c$ and $d$ are fitting parameters accounting for the cameras tilted view, and $p_x$ and $p_y$ are the camera pixel $x$ and $y$ coordinates. Parameters $a$, $b$, $c$ and $d$ were manually calibrated for each eyepiece to convert a millimeter-spaced grid placed at the mouse eye position into pixel coordinates (Fig. 4d). After pupil movements (in millimeters) relative to the eye center were calculated, eye rotations could then be estimated. Since IR glare and reflections off the Fresnel lens partially obstructed the view of the pupil, estimating eye rotations on the basis of elliptical pupil distortions as has been previously demonstrated[24] was unreliable. Instead, we estimated eye rotations on the basis of the lateral movement of the pupil center and the approximate distance between the pupil and the mouse eye center, as measured previously[25] (3.2–3.4 mm axial length and 0.35–0.4 mm anterior chamber depth, yielding ~1.3 mm from the pupil to the eye center). Pupil movements were converted to eye rotations by the following equations:

$$e_{yaw} = \arctan(p_x/p_r)$$

$$e_{pitch} = \arctan(p_y/p_r),$$

where $e_{yaw}$ and $e_{pitch}$ are the eye yaw and pitch angles, $p_x$ and $p_y$ are the pupil $x$ and $y$ coordinates relative to the eye center (rotated so the $x$ coordinate follows the horizon) and $p_r$ is the radial distance of the pupil

from the eyeball center. Eye yaw and pitch angles were mapped onto the visual field relative to their approximate optical centers located at ±70° azimuth and 10° elevation.

## Headset rotation feedback
To perform closed-loop feedback so headset rotation leads to rotation in the VR environment, an integrated sensor (6 degree-of-freedom gyroscope and accelerometer, LSM6DSOX, Adafruit) or magnetometer (LIS3MDL, Adafruit) was attached to a rotating mount supporting a MouseGoggles Duo headset. Accelerometer or magnetometer readings were measured by a Teensy 4.0 microcontroller (Teensy40, PJRC) running Arduino code to convert sensor readings into absolute headset orientation relative to straight ahead and relay these values via a Universal Serial Bus (USB) connection to the Raspberry Pi to control virtual movement.

## Animals
All animal procedures complied with relevant ethical regulations and were performed after approval by the Institutional Animal Care and Use Committee of Cornell University (protocol number 2015-0029). All mice were housed in a climate-controlled facility kept at 22° C and 40–50% humidity, under a 12 h light–dark cycle with ad libitum access to food and water. All behavioral experiments were performed during the night phase. For mouse intereye distance measurement (Extended Data Fig. 2), a variety of mouse genotypes and ages were used: C57BL/6 (three females, five males; 4–16 months old), APP$^{nl-g-f}$ heterozygotes[26] (three males, four females; 2–3 months old), TH::Cre heterozygotes (line Fl12, www.gensat.org) (one male; 16 months old) and Drd2::Cre heterozygotes (line ER44, www.gensat.org) (one male, two females; 4 months old). For two-photon calcium imaging and hippocampal electrophysiology experiments (Fig. 2), 6–9-month-old C57BL/6J male mice were used (three mice for imaging and two for electrophysiology). For virtual linear track behavioral conditioning experiments (Fig. 3a–d), ten male 2–4-month-old C57BL/6 mice were used. For looming visual stimulus behavioral experiments measuring startle reactions (Fig. 3e,f), eight male 2–7-month-old C57BL/6 mice were used. For looming visual stimulus experiments during eye and pupil tracking (Fig. 4), five male 4–5-month-old C57BL/6 mice were used.

## Surgical preparation for head-fixed behavior
Mice were anesthetized with isoflurane (5% for induction, 1% for maintenance) and placed on a feedback-controlled heating pad. Surgeries were performed on a stereotaxic apparatus where the heads of mice were fixed with two ear bars. Ointment (Puralube, Dechra) was applied to both eyes for protection. Injection of Buprenex (dose 0.05 mg kg$^{-1}$) was given for analgesia. Lidocaine (2.5 mg kg$^{-1}$) was administered to the scalp after being disinfected by 75% ethanol and povidone–iodine. A small incision (~12–15 mm) along the sagittal line of the skull was made to expose a section of the skull sufficiently large to place a custom-designed titanium head plate. The head was rotated so that the bregma and lambda features of the skull were level. The surface of the skull was gently scratched by a scalpel to remove the periosteum. After the skull was completely dry, a thick layer of Metabond (Parkell) was applied to cover the skull surface. A titanium head plate was mounted on top of the Metabond and aligned with the surface of the skull and position of the eyes. The head plate was further secured by an additional layer of Metabond. Postoperative ketoprofen and dexamethasone were administered subcutaneously, and the mouse was returned to its home cage on a heating pad for recovery. All behavior tests were performed at least 1 week after surgery.

## Surgical preparation for calcium imaging
Mice underwent surgical procedures for head-fixed behavior with modifications to accommodate a viral injection. First, a 3 mm craniotomy was made above V1 (anterior-posterior 3 mm, medial-lateral

2.5 mm from Bregma, centerline) on the right hemisphere. A 50 nl bolus of AAV9-Syn-GCaMP6s (Addgene) diluted to $10^{12}$ vg ml$^{-1}$ was injected to the target V1 layer 2/3 (dorsal-ventral −0.2 mm from the brain surface). A 3 mm glass window then replaced the hole in the skull, and the titanium head plate was secured to the skull with Metabond. Postoperative ketoprofen and dexamethasone were administered subcutaneously, and the mouse was allowed to fully recover in a cage on a heating pad. Four weeks were allowed for viral expression before imaging.

## Surgical preparation for electrophysiology
Mice underwent surgical procedures for head-fixed behavior with modifications to accommodate a chronic electrode implant. First, a craniotomy was made above the dorsal CA1 (anterior-posterior 1.95 mm, medial-lateral 1.5 mm from Bregma, centerline), and a burr hole was made in the contralateral occipital plate for the placement of a ground screw. A stainless-steel wire was soldered to the ground screw and threaded through the head plate, which was then secured to the skull with Metabond. A 64-channel single-shank silicon probe (NeuroNexus) was adhered to a metal moveable micro drive (R2Drive, 3Dneuro) to allow vertical movement of the probe after implantation. The probe was implanted above the dorsal CA1 (dorsal-ventral −1.1 mm from brain surface), and the craniotomy was sealed with a silicone elastomer (DOWSIL 3-4680, Dow Chemical). Copper mesh was fixed to the Metabond that surrounded the micro drive and formed a cap. Ground and reference wires were soldered to the copper mesh to reduce environmental electrical noise. While the mouse was in the home cage, the copper mesh was covered with an elastic wrap to prevent debris from entering the cap.

## Two-photon microscopy
Two-photon calcium imaging was performed using a Ti:Sapphire laser (Coherent Vision S Chameleon; 80 MHz repetition rate, 75 fs pulse duration) at 920 nm to excite the GCaMP6s calcium indicator, with ~35 mW power at the sample. Imaging signals were acquired using ScanImage[27] software (SI2022) into separate blue and green color channels (separated by a 488 nm long-pass dichroic; channel 1 using a 510/84 (center wavelength/bandwidth, both in nanometers) bandpass filter and channel 2 using 517/65). A transistor-transistor logic (TTL) signal from the monocular display was acquired into an additional unused imaging channel for synchronizing imaging with visual stimuli. The 256 × 256 pixel image frames were acquired at 3.41 Hz.

## Monocular visual stimulation
Visual stimulation experiments were performed with anesthetized, head-fixed mice by positioning a MouseGoggles Mono display at the mouse's left eye for contralateral two-photon imaging in the right hemisphere. The display was oriented to 45° azimuth and 0° elevation respective to the long axis of the mouse. To measure display light contamination, a maximum-brightness blue square (covering a 66°-wide region of the visual field) was flickered at 0.5 Hz for five repetitions, first by the monocular display, then by a flat LED monitor (ROADOM 10.1' Raspberry Pi Screen, Amazon) positioned 10 cm from the mouse eye and oriented at 70° azimuth and 0° elevation, with the cranial window either unblocked or blocked with a circular cut piece of black masking tape (T743-2.0, Thorlabs) (Extended Data Fig. 5). To measure V1 neuron visual stimulus encoding, neurons with RFs in the monocular display center were located by presenting a drifting grating stimulus in a small square region in the center of the display (24-pixel/15.3°-wide square and 24 pixel/15.3° SF) once every 6 s, cycling through four directions (right, down, left and up) until V1 L2/3 neurons excited by the stimulus were found through the live view of the fluorescence microscope. In this position, a single visual stimulation protocol was performed with three mice. All stimuli were blue square-wave gratings shown at 100% contrast, 1 Hz temporal frequency (TF) and 20 pixel/12.7° SF unless otherwise noted. First, RF mapping was performed by presenting a four-direction bar sweep stimulus (right, down, left and up; 0.5 s per direction, 2 s total) in one location at a time in a 5 × 5 grid at the center of the display. Each segment of the grid was a 12-pixel/7.6°-wide square, where 6 × 12 pixel/3.8 × 7.6° bright bar sweeps were shown. Stimuli were presented in the 25 segments one at a time in a random order with 6 s between stimuli, for a total of five repetitions at each location. Next, orientation and direction tuning were measured by presenting drifting gratings at 12 angles (0–330°, in 30° increments) in a random order, at a 40 pixel/25.5° SF in an 80-pixel/51°-wide square that rotated on the basis of the angle of the grating. Each stimulus was 1 s in duration with 6 s between stimuli, for five repetitions. Finally, SF, TF and contrast tuning were measured using unidirectional (rightward) drifting gratings in an 80-pixel/51°-wide square region; only a single direction was used to reduce the number of stimuli and the duration of the experiment. The stimulus set consisted of the default grating stimulus (100% contrast, 1 Hz TF, 12.7° SF) varied across five SFs (4, 10, 20 40 and 80 pixels; 2.5°, 6.4°, 12.7°, 25.5° and 51°, respectively), six TFs (0.5, 1, 2, 4, 8, and 12 Hz) and six contrast values (5-bit bright/dark bar values of 15/15, 18/12, 21/9, 24/6, 27/3 and 30/0—for contrast values of 0%, 20%, 40%, 60%, 80% and 100%, respectively), for a total of 15 unique stimuli. Each stimulus was 2 s in duration with 6 s between stimuli, for five repetitions.

## Calcium imaging analysis
Scanimage Tiff files were processed through suite2p[28] (v0.10.3) for motion stabilization, active region of interest segmentation and spike inference, followed by custom Matlab scripts for analysis and plotting. Segmented cells were manually screened for accurate classification, resulting in 410 cells pooled from 3 mice (142, 112 and 156 cells from mouse 1, 2 and 3, respectively). For each cell, time-series vectors of the extracted fluorescence and inferred spikes were aligned for each stimulus repetition to calculate the average stimulus response. Activity was quantified at baseline and during each stimulus from the mean of all inferred spikes during baseline frames (2 s preceding each stimulus) and during the stimulus presentation (1 or 2 s in duration depending on the stimulus). RF size was estimated similarly to previous methods[9]. In brief, the mean response to the four-direction stimuli presented in a 5 × 5 grid was fit with a 2D Gaussian function using lsqcurvefit(@D2GaussFunctionRot) in Matlab, estimating an ellipse with two independent widths for the major and minor axes. The RF size was calculated by averaging the half width at half maximum of the major and minor axes. Only cells well fit to the 2D Gaussian were included (resnorms <0.25; 341 cells). Normalized SF tuning curves were estimated similarly to previous methods[10] by fitting each cell's responses with a log-Gaussian function:

$$R(\text{SF}) = e^{-\frac{(\log_2 \text{SF} - \log_2 \text{SF}_{\text{pref}})}{2\sigma^2}},$$

where SF is the spatial frequency, $\text{SF}_{\text{pref}}$ is the preferred spatial frequency and $\sigma$ is a fitting parameter describing the width of the curve. Only cells well fit to the function were included (adjusted $R^2 > 0.8$; 124 cells). Normalized contrast tuning curves were estimated similarly to previous methods[11] by fitting each cell's responses with a Naka–Rushton function:

$$R(c) = \frac{c^n}{c^n + c_{50}{}^n},$$

where $c$ is the contrast, $c_{50}$ is the semisaturation contrast and $n$ is a fitting exponent describing the sharpness of the curve. Only cells well fit to the function were included (adjusted $R^2 > 0.8$, 202 cells).

## Spherical treadmill and lick port
The spherical treadmill system was built on the basis of an existing design[1]. A 20-cm-diameter Styrofoam ball was suspended by compressed air. Locomotion of mice was tracked through ball movement by two optical flow sensors (ADNS-3080 Optical Flow Sensor APM2.6)

mounted on the bottom and side of the ball. The optical sensors sent the ball motion to an Arduino Due via SPI which was processed by a custom script (https://github.com/Lauszus/ADNS3080) and relayed via a USB connection to a Raspberry Pi or PC. The roll, pitch and yaw movements of the ball were transformed to drive the corresponding animal movements in the VR environment. Velocity gain was calibrated to ensure a one-to-one correspondence between the distances traveled in the virtual environment and on the surface of the ball. A 1.83-mm-diameter stainless-steel lick port along with a customized capacitive sensor[29] was used to deliver water rewards and measure licking behavior. Water was delivered through a solenoid valve (SSZ02040672P0010, American Science Surplus) operated by a Teensy 4.0 microcontroller (Teensy40, PJRC) and relay (4409, Adafruit). The microcontroller received valve open commands from and transmitted lick detections to the Raspberry Pi over USB using Xinput (https://github.com/dmadison/ArduinoXInput).

### Linear treadmill

The linear treadmill was modified from an existing design (linear treadmill with encoder, Labmaker). Custom 3D-printed wall plates were used to accommodate a larger head mount, and custom code was uploaded to the treadmill microcontroller to convert treadmill motion into emulated computer mouse *y* movements, relayed via a USB connection to the Raspberry Pi to control virtual movement. Similarly to the spherical treadmill, velocity gain was calibrated to ensure a one-to-one correspondence between the distances traveled in the virtual environment and on the surface of the treadmill.

### Head-fixed behavioral tests

Before the start of behavioral tests, all mice were habituated in the room where the experiment would be performed for at least 1 day, followed by at least 5 days of habituation on the spherical or linear treadmill (without a VR system attached). On each treadmill training day, mice were head-fixed on a custom-designed holder via the mounted head plate, and the head was positioned on the center of the spherical treadmill or positioned on the linear treadmill so that the body was contained by the treadmill walls. After room and treadmill habituation, mice were then habituated to the MouseGoggles Duo or MouseGoggles Eyetrack headset by positioning the headset to the mouse eyes using sliding optical posts so that both eyes were approximately positioned at the center of the eyepieces, typically 0.5–1 mm from the lens surface. A gray image was then shown on the VR display for 10 min before VR experiments were started.

### Hippocampal electrophysiology and analysis

Recordings were conducted using the Intan RHD2000 interface board or Intan Recording Controller, sampled at 30 kHz. Amplification and digitization were done on the head stage. Data were visualized with Neurosuite software (Neuroscope). Mice concurrently underwent neural activity screening and head-fixed behavior habituation. For screening, activity from each amplifier channel was monitored while a mouse foraged for sugar pellets in an open field (30 cm × 30 cm × 12 cm), and the electrode was lowered (<125 μm per day) until area CA1 layers were visible, identified by physiological features of increased unit activity and local field potential ripples[30]. The mouse's virtual position on the ball was synchronized with electrophysiological data using a TTL pulse from the Teensy microcontroller connected to the Raspberry Pi.

### Spike sorting, unit identification and encoding of virtual position

Electrophysiology data were analyzed with custom Python code (https://github.com/lolaBerkowitz/SNLab_ephys) using the Nelpy python package (https://github.com/nelpy/nelpy). Spike sorting was performed semiautomatically with KiloSort (https://github.com/cortex-lab/KiloSort), followed by manual curation using the software

Phy (github.com/kwikteam/phy) and custom-designed plugins (https://github.com/petersenpeter/phy-plugins). Identified units were assessed by manual inspection of auto-correlograms, waveforms, waveform distribution in space and PCA metrics. Units with high contamination in the first 2 ms of the auto-correlogram or with visible noise clusters were discarded. Spatial tuning curves were created by binning spike data and the mouse's virtual position into 3 cm bins. Raw spike and occupancy maps were smoothed using a Gaussian kernel (3 cm s.d.). Only spike data from when the animal's velocity was greater than 5 cm s$^{-1}$ was used. A spatial information content score[31] was calculated for each cell by the following definition:

$$\text{SI} = \sum_{i=1}^{N} P_i x \frac{\lambda_i}{\lambda} \log_2 \frac{\lambda_i}{\lambda},$$

where the virtual environment is divided into $N$ spatial bins, $P_i$ is the occupancy of bin $i$, $\lambda_i$ is the mean firing rate for bin $i$ and $\lambda$ is the overall mean firing rate of the cell. A surrogate set of information content scores was created by shuffling the position coordinates 500 times and computing the spatial information content score for the resulting tuning curves at each shuffle. A cell was defined as a place cell if the observed information content score was greater than the 95th percentile of shuffled scores and if the cell's peak rate was at least 1 Hz. For cells that met these requirements for spatial information content, field detection was performed using the find_fields function in neuro_py (https://github.com/ryanharvey1/neuro_py). In brief, a field was defined as an area that encompassed at least 30% of a local peak of the ratemap. Place fields were at minimum 10 virtual cm and at most 80 virtual cm, or 7–53% of the virtual linear track (150 virtual cm length).

### Linear track place learning

A virtual linear track was designed for the MouseGoggles Duo headset using the Godot video game engine (https://godotengine.org/). The track was 1.5 m long and 6 cm wide, with 5-cm-high walls that were divided into three equal-length, visually distinct wall sections: (1) black and green vertical stripes, (2) black and white spots and (3) black and green horizontal stripes. In addition, a tall black tower was located at 0.9 m (track start 0 m, track end 1.5 m) to provide a more distal cue of location. The virtual location of the mouse began at 0.04 m and oriented at 0°, looking straight down the track; mice were constrained to locations within the track that were at least 4 cm from the nearest wall to prevent camera views clipping through the walls, limiting the total habitable length of the track to 1.42 m. The spherical treadmill pitch controlled forward/backward walking, while mouse heading was maintained at 0° to keep mice traversing down the track. Liquid rewards were given through the lick port at a specific location along the track to condition licking behavior at that location over time; rewards were given at 0.5 m for mice 1–5 (cohort A) and at 1.0 m for mice 6–10 (cohort B). Three days before training, mice were provided with 1.2 ml of water daily and their body weight was continuously monitored, with additional water supplementation administered to maintain their body weight above 85% of their pretraining weight. After habituation for head-fixed behavior, all mice underwent a 5-day linear track place-learning protocol with the following parameters:

- Days 1–2: liquid reward is automatically delivered when the mouse reaches the reward location.
- Day 3: for the first three trials, liquid reward is automatically given. For trials 4+, the mouse must first lick in the reward zone (no farther than 0.25 m away from the reward location) before a reward is delivered at the mouse's location of licking.
- Days 4–5: similar to day 3 (trials 1–3 guarantee reward; trials 4+ require licks), with a random 20% of trials unrewarded (probe trials).

Mice performed one session of track traversals per day, where each session consisted of 40 laps down the track. Once mice reached the end of the track (located at 1.46 m), the traversal finished, and the mice were teleported back to the beginning to start a new lap. If mice did not reach the track end within 60 s, the trial data were discarded but still counted toward the 40-lap session limit. Licks were detected by the rising edge of the lick sensor and were recorded alongside mouse position during each traversal. All licking data was binned by location into 5-cm-wide bins, while the first and last bin were excluded due the mouse's constrained position away from the walls. Lick rates were calculated by dividing the number of licks in each binned position by the time spent at that position. 'Post-reward' licks were defined as a series of licks that quickly followed a reward delivery (starting within 3 s of a delivered reward) and continued until the lick rate dropped below 1 lick s$^{-1}$. All licks occurring at other times than after a reward delivery were defined as 'exploratory licks'. Reward and control zones were defined as regions spanning ±0.25 m (ten total position bins) from the rewarded location. The fraction of exploratory licks in the reward and control zones were calculated by dividing the total number of licks in each zone by the total licks in all 28 habitable bins. Chance-level zone licking was calculated by dividing the size of the zones in bins by the total habitable zone of the track (10/28 = 35.71%). During days 4–5, data were subdivided into rewarded versus unrewarded 'probe' trials, where probe trials contain no post-reward licks. Statistically significant differences in the proportion of licks in reward versus control zones during probe trials was calculated using the two-tailed Mann–Whitney $U$ test (ranksum function in Matlab), comparing the mean lick proportions of each mouse with trials pooled from days 4 and 5.

## Projector-based VR system
To compare the VR headset with a traditional panoramic display, a VR environment was generated by the Unity video game engine and projected onto a custom-built conical rear-projection screen (Stewart FilmScreen 150) surrounding the mouse using two projectors (Optoma HD141X Full 3D 1080p 3000 Lumen DLP). The projection screen covered 260° in azimuth of the mouse's visual field and spanned an elevation of 92° (−28° to 64°) with a circular hole at the top to accommodate a microscope objective (Extended Data Fig. 7).

## Loom–startle experiment and analysis
Head-fixed mice walking on the spherical treadmill were recorded using an HD webcam (NexiGo N980P, 1080p 60 fps). Mice were shown looming visual stimuli with the MouseGoggles Duo headset, appearing as a dark circular object in the sky 45° in elevation and 45° to either the left or right from straight ahead, beginning 20 m away and approaching at 25 m s$^{-1}$ until disappearing at 0.6 m away. Left or right looms were displayed pseudorandomly, separated by 10 s, for ten repetitions (five left and five right). Video clips of mice during each looming stimulus presentation (from 5 s before the loom to 5 s after) were created and evaluated by two independent scorers who were blind to the goals of the experiment, although they could not be completely blinded to the experimental condition (headset versus projector) owing to the nature of the recorded videos. Both scorers were given the following instructions for determining startle responses (and other behaviors) from clips:

"In each 10 s clip, look for a behavioral reaction to the loom stimulus. The loom is a dark spot that appears on the screen, grows exponentially in size, then disappears, the full process taking ~0.8 s. The loom may be easy or hard to see based on the display used and the starting position of the loom, but will always be visible when it reaches its max size and disappears.

[For each clip], write down your confidence level (0–3) in seeing that reaction in each clip. Except for grooming, these reactions should be something the mouse was not doing before the loom, but started doing during or immediately at the end of the loom. For the grooming reaction, write down whether the mouse was grooming during the stimulus, even if it was grooming before it as well.

Possible reactions:
- Startle: burst of movement, jump or kick
- Tense up: back arches, tailbone or tail curling under
- Stop: stops, from a moving state
- Run: starts running, from a stopped or slowly walking state
- Turn: rear end of its body swings to the side
- Grooming: uses its paws to wipe at mouth/whiskers/eyes

*Confidence scores:*
- 0: reaction did not happen
- 1: reaction possible happened
- 2: reaction probably happened
- 3: reaction definitely happened"

Due to the startle and tense up reactions being difficult for the scorers to differentiate, the individual confidence scores for these two behaviors were combined into a single reaction score, where the larger of the two became the new score. To classify responses for each mouse, repetition and experimental condition, the average of the two scores (one from each scorer) was taken, where average scores of 1.5 or greater were classified as a startle response to the looming stimulus. The proportion of startle responses was calculated by dividing the number of startle responses by the number of observations at each repetition and was fit with an exponential decay function with offset:

$$R(r) = R_1 e^{-\lambda(r-1)} + b,$$

where $r$ is the repetition number, $R_1$ is the startle response proportion at $r = 1$, $\lambda$ is the decay rate constant and $b$ is the offset. The experiment was initially performed with two mice where startle responses were first observed in the VR headset, after which a second cohort of mice was tested with both the VR headset and the projector-based system. For mice that began with the headset VR (4/6 mice), 10 days elapsed before testing with the projector to attempt to restore the novelty of the looming stimulus. For mice that began with the projector VR, only 1 day elapsed before testing with the headset. Neither of the two 'projector-first' mice was startled in projector VR, but both were startled in headset VR.

## Loom–eye-tracking experiment and analysis
Head-fixed mice walking on the linear treadmill were presented with looming visual stimuli similar to the loom–startle experiment, using the MouseGoggles EyeTrack headset. Fifteen repetitions of the looming stimulus were presented in five sets of three conditions: a looming object approaching from 45° right, 45° left or center. The loom was 45° in elevation for all conditions, and the visual scene was blacked out above 64° elevation to match the vertical extent of the projector VR system. Treadmill velocity and the timing of looming stimulus presentation was logged by the game engine. An eye-tracking video for each eye was acquired independently and synchronized to the looming stimuli afterward using the blue channel of the eye-tracking video, which acquires a partial view of the VR display through the hot mirror so that looming stimuli can be observed (alongside eye tracking in the red channel). Post-loom walking speed change was calculated from the walking speed after the loom (average velocity during 2 s from loom onset), relative to the baseline walking speed (average velocity during 2 s before stimulus onset). The post-loom eye pitch angle change and pupil diameter change were calculated similarly, except that the average pupil diameter change was calculated during the 3 s after the loom onset owing to the relatively slower pupil response we observed. Post-loom walking speed change, eye pitch angle change and pupil diameter change were then averaged across each set of three looming conditions (left, right and center) for each mouse individually.

Statistically significant trends of these three measurements by repetition number of the looming stimuli were determined by Cuzick's trend test in Matlab (https://github.com/dnafinder/cuzick). An average time-series response of walking speed and eye pitch change was calculated by averaging the response across all looming repetitions for each mouse individually, followed by calculating the average and s.d. of the response across mice. Since the pupil diameter response habituated with increased loom repetition, an average time-series response of pupil diameter change was calculated using only the first set of three looming stimuli.

### Reporting summary

Further information on research design is available in the Nature Portfolio Reporting Summary linked to this article.

### Data availability

Datasets are deposited in the figshare database at https://doi.org/10.6084/m9.figshare.24039021.v4 (ref. 32).

### Code availability

Code and hardware designs are available upon request.

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

### Acknowledgements

This project was supported by the Cornell Neurotech Mong Family Fellowship program (M.I. and H.C.); the BrightFocus Foundation Alzheimer's disease fellowship program (grant no. A2023006F, M.I.); the Brain and Behavior Research Foundation (grant no. 139526, I.E.) and the National Institutes of Health (grant no. R01 AG081931, C.B.S.). We thank A. Oliva, A. Fernandez-Ruiz and W. Tang for their comments on the manuscript; A. Grosmark, A. Kaye, S. Staszko and E. Krishnamurthy for their feedback to improve the reproducibility of the method; and A. Huang and A. Wulf for mouse behavior video scoring.

### Author contributions

M.I. conceived and built the monocular display and binocular VR headset, with H.C., I.E. and C.B.S. providing feedback. H.C. and M.I. designed the eye-tracking hardware and analysis pipeline. I.E. designed and built the comparative panoramic VR system. M.I., H.C., I.E. and C.B.S. jointly designed all experiments. H.C. prepared animals and performed linear track place learning and looming behavioral experiments. L.B. prepared mice and conducted and analyzed data for linear track behavior assays during electrophysiological recording. R.Z. prepared mice and conducted calcium imaging experiments. M.I. analyzed behavioral, eye-tracking and calcium imaging data. Y.P. developed software communication protocols for the monocular display. D.H. tested and validated the Godot game engine for the binocular headset. I.E. and C.B.S. provided guidance in all aspects of the work. M.I. wrote the paper with contributions from all authors.

### Competing interests

The authors declare no competing interests.

### Additional information

**Extended data** is available for this paper at https://doi.org/10.1038/s41592-024-02540-y.

**Correspondence and requests for materials** should be addressed to Matthew Isaacson.

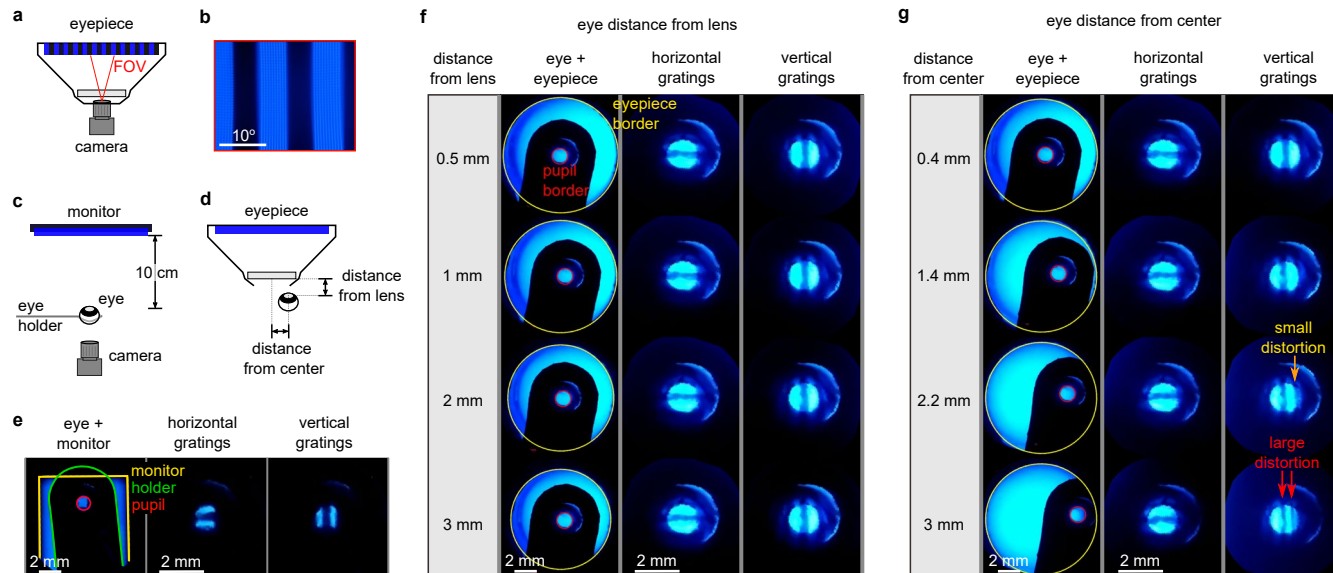

**Extended Data Fig. 1 | Display projection through an enucleated mouse eye.**
**a**, Schematic layout of a MouseGoggles eyepiece with a mini camera set to infinite focal distance, positioned 1 mm from the eyepiece lens center, with a field of view (FOV) centered on the display. **b**, Image of the eyepiece display produced from the imaging setup in (**a**). **c**, Layout of an enucleated mouse eye positioned on a 3D printed holder 10 cm below a traditional monitor, with a mini camera positioned below the eye. **d**, Layout of an enucleated mouse eye positioned below a MouseGoggles eyepiece, with a variable eye position relative to the lens center. **e**, Images produced from the imaging setup in (**c**), with views of a uniform brightness image (left), horizontal gratings (middle), and vertical gratings (right). Images of the eye during horizontal (middle) and vertical (right) gratings

are at 2x zoom relative to the image with uniform brightness (left).
**f**, Images produced from the imaging setup in (**d**), with eye distance-from-lens values of 0.5, 1, 2, and 3 mm (top to bottom), with views of a uniform image (left), horizontal gratings (middle), and vertical gratings (right). **g**, Images produced from the imaging setup in (**d**), with eye distance-from-center values of 0.4, 1.4, 2.2, and 3 mm (top to bottom), with views of a uniform image (left), horizontal gratings (middle), and vertical gratings (right), and with small and large distortions marked for the 2.2 mm and 3 mm positions. All images in this figure were taken using the same enucleated mouse eye, with similar results reproduced using a second eye.

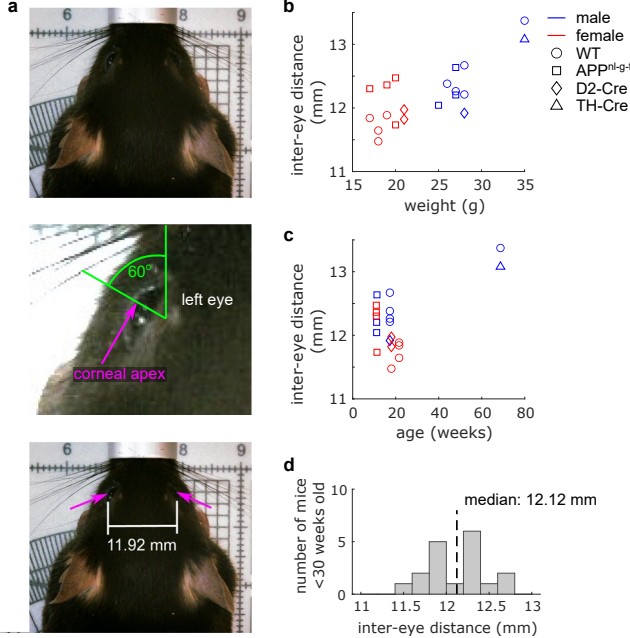

**Extended Data Fig. 2 | Mouse inter-eye distance. a**, Views of an anesthetized mouse from above for measuring distance between corneal apexes. **b**, Scatterplot of eye distance measurement as a function of mouse weight, for both male and female mice of different genotypes. **c**, Scatterplot of the data in (**b**) plotted alternatively as a function of mouse age. **d**, Histogram of inter-eye distances of all mice younger than 30 weeks (n = 18 mice, 11–22 weeks).

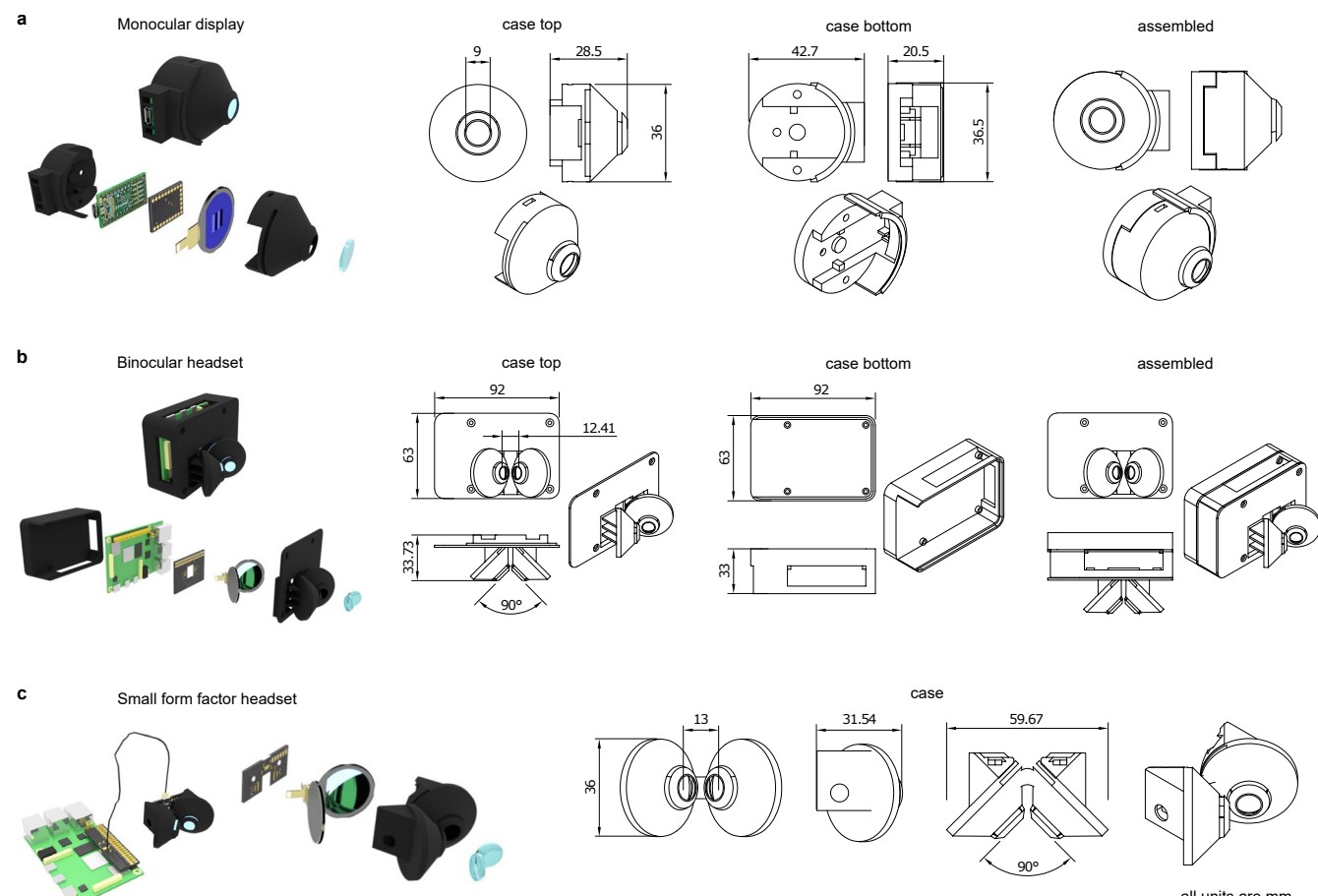

**Extended Data Fig. 3 | Designs of monocular and binocular display enclosures. a**, (left) 3D renders of assembled and exploded views of the monocular display, MouseGoggles Mono (version 1.0). (right) CAD designs and overall dimensions (in mm) of the top and bottom halves of the display case as well as the fully assembled case. **b**, (left) 3D renders and (right) CAD designs of the binocular headset, MouseGoggles Duo (version 1.0). **c**, (left) 3D renders and (right) CAD designs of the smaller form factor binocular headset (MouseGoggles Duo version 1.1), connecting to a raspberry Pi with a SPI cable. All units are in mm.

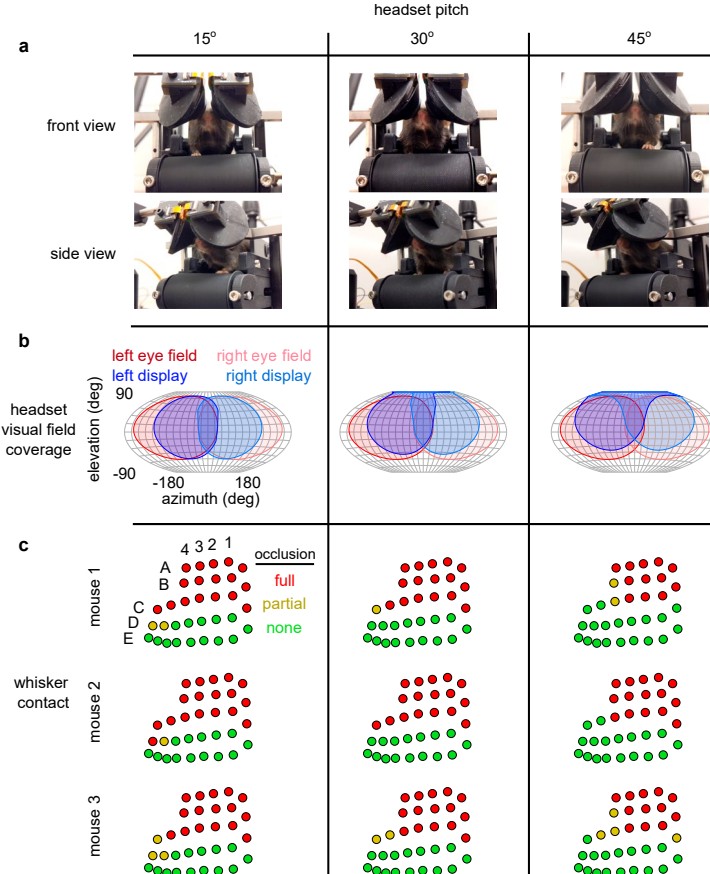

**Extended Data Fig. 4 | Whisker occlusion by headset pitch. a**, Front (top) and side (bottom) views of a head-fixed mouse on a linear treadmill, positioned with a MouseGoggles Duo (version 1.1) headset at three pitch angles: 15 deg (left), 30 deg (middle), and 45 deg (right). **b**, Estimated visual field coverage of the headset for the three pitch angles in (**a**). **c**, Map of whiskers which make constant/full contact, temporary/partial contact, or no contact with the headset at the three pitch angles in (**a**), measured for three different mice.

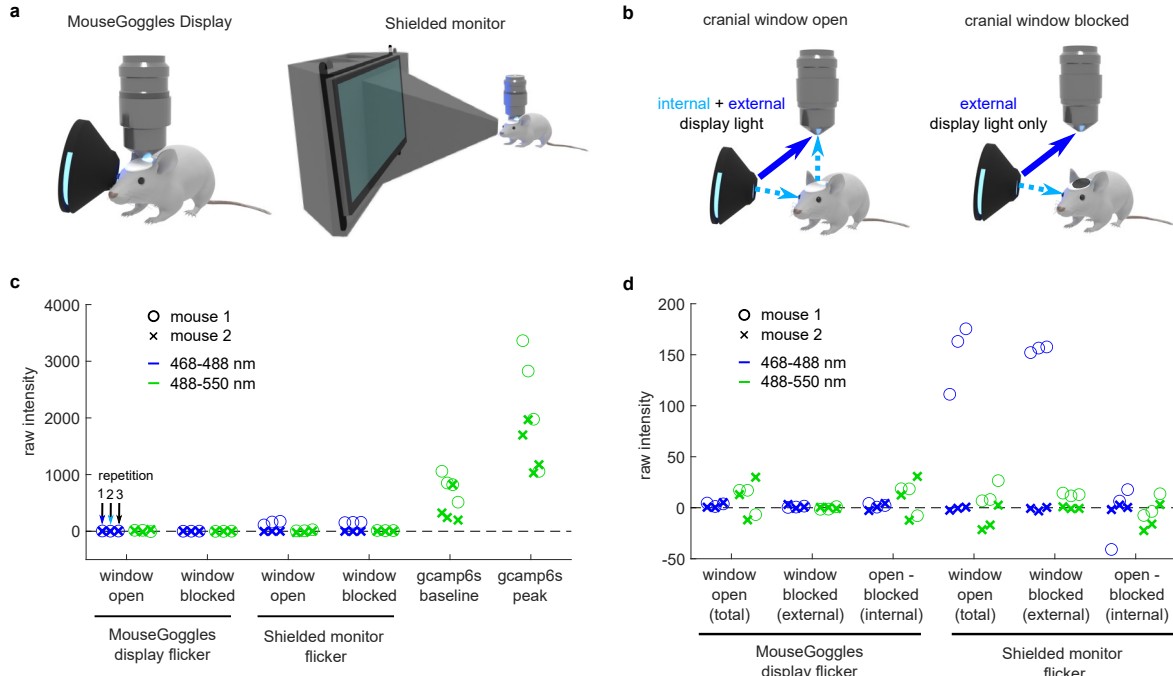

**Extended Data Fig. 5 | Display light pollution measurements for two-photon imaging. a**, 3D renders of a two-photon calcium imaging setup for head-fixed mice with visual stimulation from either a MouseGoggles Mono eyepiece (left) or a fully-shielded flat monitor (right). **b**, Diagram of brain imaging during visual stimulation with the cranial window unblocked (left) or blocked (right), with pathways indicating both external light detection and internal light detection (light travelling through the pupil and scattering through the brain). **c**, Scatterplot of raw light intensity increase during 3 repetitions of a maximum brightness blue image flicker (relative to black image baseline), measured in blue (468-488) and green (488–550 nm) imaging channels, from 2 mice. Measurements are compared between a MouseGoggles display and Shielded monitor display, with either the cranial window open (total stray light) or blocked (external pathway stray light only), as well as the raw intensity measurements from GCaMP6s baseline and peak fluorescence during a typical calcium imaging experiment (4 representative GCaMP6s-labeled cells each from 2 mice). **d**, Scatterplot of data, reproduced from panel (**c**), zoomed in on the y-axis to show small intensity measurements. Additional columns added for the internal pathway stray light measurements (that is 'external' subtracted from 'total').

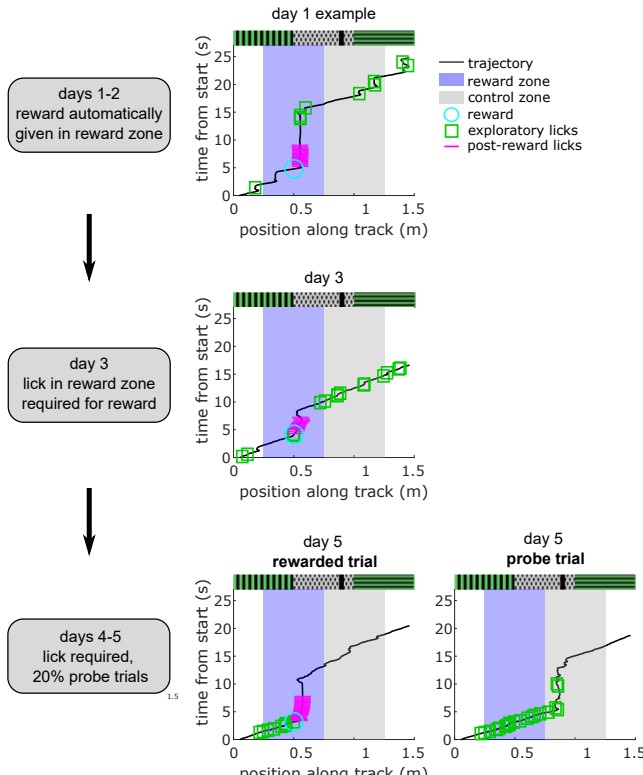

**Extended Data Fig. 6 | Rewarded linear track protocol and example trajectories.** (left) Description of the 5-day training protocol for virtual linear track spatial-learning. (right) Example mouse trajectories during training days 1, 3, and 5, with detected licks and delivered liquid reward overlayed on the trajectory. Two example trajectories are shown for day 5, including one rewarded trial and one probe trial.

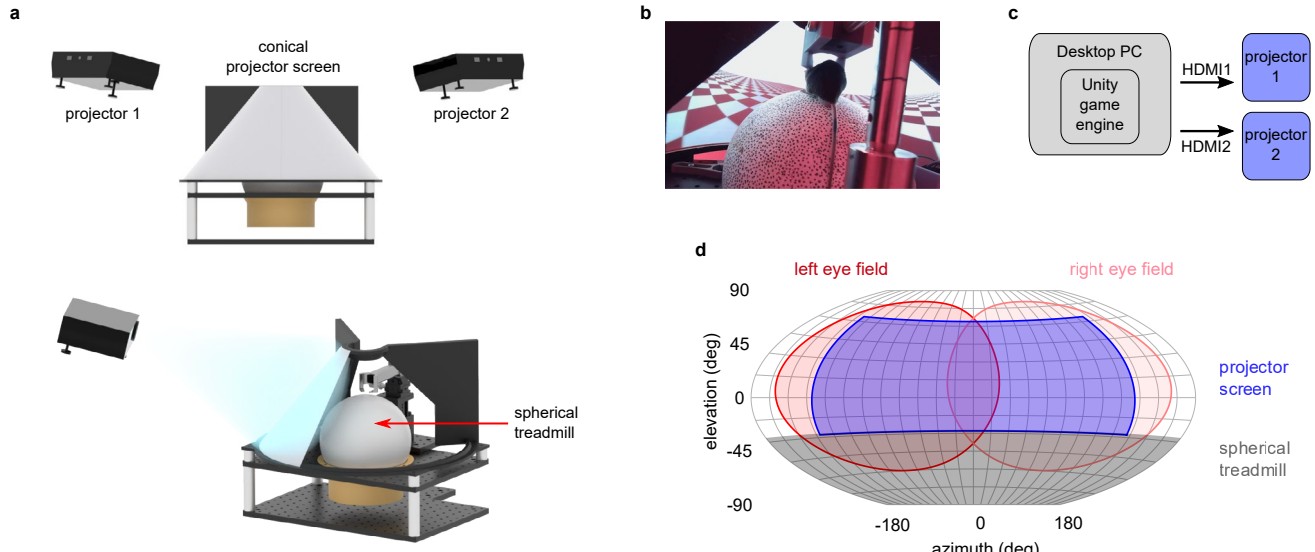

**Extended Data Fig. 7 | Projector-based VR system for looming reaction comparison. a**, 3D renders of the custom projector-based VR system composed of two HD projectors, a high FOV conical screen, and spherical treadmill. **b**, Image of a head-fixed mouse running on the spherical treadmill inside the projector-based VR system. **c**, Communication diagram of the Desktop PC and Unity game engine system for rendering 3D environments onto the dual-projector display. **d**, Winkel tripel projection of the mouse's estimated visual field overlayed with the estimated visual field coverage of the projector screen and spherical treadmill.

# Reporting Summary

## Statistics

For all statistical analyses, confirm that the following items are present in the figure legend, table legend, main text, or Methods section.

| n/a | Confirmed | |
|---|---|---|
| ☐ | ☒ | The exact sample size (*n*) for each experimental group/condition, given as a discrete number and unit of measurement |
| ☐ | ☒ | A statement on whether measurements were taken from distinct samples or whether the same sample was measured repeatedly |
| ☐ | ☒ | The statistical test(s) used AND whether they are one- or two-sided *Only common tests should be described solely by name; describe more complex techniques in the Methods section.* |
| ☒ | ☐ | A description of all covariates tested |
| ☐ | ☒ | A description of any assumptions or corrections, such as tests of normality and adjustment for multiple comparisons |
| ☐ | ☒ | A full description of the statistical parameters including central tendency (e.g. means) or other basic estimates (e.g. regression coefficient) AND variation (e.g. standard deviation) or associated estimates of uncertainty (e.g. confidence intervals) |
| ☐ | ☒ | For null hypothesis testing, the test statistic (e.g. *F*, *t*, *r*) with confidence intervals, effect sizes, degrees of freedom and *P* value noted *Give P values as exact values whenever suitable.* |
| ☒ | ☐ | For Bayesian analysis, information on the choice of priors and Markov chain Monte Carlo settings |
| ☒ | ☐ | For hierarchical and complex designs, identification of the appropriate level for tests and full reporting of outcomes |
| ☒ | ☐ | Estimates of effect sizes (e.g. Cohen's *d*, Pearson's *r*), indicating how they were calculated |

*Our web collection on statistics for biologists contains articles on many of the points above.*

## Software and code

Policy information about availability of computer code

**Data collection**

All custom code used for data collection is available upon request, with the following open and commercial software dependencies listed here:
Raspberry Pi OS 32-bit (VR display operating system)
Godot 3.2.3.stable.flathub (VR experiment creation and rendering)
Arduino 1.8.15 (monocular display control; VR display I/O communication)
Teensyduino 1.57 (monocular display control; VR display I/O communication)
Matlab 2022b (imaging acquisition and visual stimulus control)
ScanImage SI2022 (Imaging acquisition)
https://github.com/adafruit/Adafruit-GFX-Library (visual stimulus generation)
https://github.com/juj/fbcp-ili9341 (SPI display driver)
https://github.com/Lauszus/ADNS308 (spherical treadmill motion tracking)
https://github.com/dmadison/ArduinoXInput (VR display I/O communication)

**Data analysis**

All custom code used for data analysis is available upon request, with the following open and commercial software dependencies listed here:
Matlab 2022b (general data analysis)
Python 2.8.8 (general data analysis)
suite2p v0.10.3 (suite2p.org) (calcium imaging analysis)
https://github.com/lolaBerkowitz/SNLab_ephys (electrophysiology analysis)
https://github.com/nelpy/nelpy (electrophysiology analysis)
https://github.com/ryanharvey1/neuro_py (electrophysiology analysis)

For manuscripts utilizing custom algorithms or software that are central to the research but not yet described in published literature, software must be made available to editors and reviewers. We strongly encourage code deposition in a community repository (e.g. GitHub). See the Nature Portfolio guidelines for submitting code & software for further information.

## Data

Policy information about availability of data

All manuscripts must include a data availability statement. This statement should provide the following information, where applicable:
- Accession codes, unique identifiers, or web links for publicly available datasets
- A description of any restrictions on data availability
- For clinical datasets or third party data, please ensure that the statement adheres to our policy

All data used in this manuscript is available upon request. Large datasets are deposited in the Figshare database at https://doi.org/10.6084/m9.figshare.24039021.v4.

## Human research participants

Policy information about studies involving human research participants and Sex and Gender in Research.

| | |
|---|---|
| Reporting on sex and gender | NA |
| Population characteristics | NA |
| Recruitment | NA |
| Ethics oversight | NA |

Note that full information on the approval of the study protocol must also be provided in the manuscript.

# Field-specific reporting

Please select the one below that is the best fit for your research. If you are not sure, read the appropriate sections before making your selection.

☒ Life sciences ☐ Behavioural & social sciences ☐ Ecological, evolutionary & environmental sciences

For a reference copy of the document with all sections, see nature.com/documents/nr-reporting-summary-flat.pdf

# Life sciences study design

All studies must disclose on these points even when the disclosure is negative.

| | |
|---|---|
| Sample size | No sample size calculation was performed. Sample sizes were based on previous scientific literature in the respective fields of study (Niell et al, J Neurosci 2008; Tan et al, Sci Rep 2015; Busse et al, J Neurosci 2011; Dombeck et al, Nat Neurosci 2010) using a minimum number of mice to successfully replicate previous findings. |
| Data exclusions | For linear track place learning experiments where 40 track traversal took place per session, some session exceeded 40 laps due to program glitches -- these extra trials were discarded. For all other experiments, no data was excluded from analysis. |
| Replication | For electrophysiology experiments, replication was performed with 2 mice. For all other experiments, replication was performed with at least 3 mice. All replication attempts were successful. For the looming visual stimulus experiment where a novel startle response was observed with an initial dataset of 2 mice, additional replication was attempted (and successful) with an additional 6 mice. |
| Randomization | For the looming visual stimulus experiment, mice were randomly allocated to either headset-first or projector-first experimental conditions. For linear track place learning experiment, mice were randomly allocated to the reward zone A vs reward zone B conditions. For all other experiments, randomization is not relevant for our study as they did not have multiple experimental groups. |
| Blinding | For mouse behavioral scoring of looming visual stimulus experiment, 2 independent scorers were blinded where possible; scorers were blinded to the purpose and details of the experiment, though they could not be completely blinded to the experimental condition (headset vs projector) due to the nature of the recorded videos which include equipment specific to each condition. All other data collection was performed through objective and unbiased automated data collection pipelines where blinding is not relevant. |

# Reporting for specific materials, systems and methods

We require information from authors about some types of materials, experimental systems and methods used in many studies. Here, indicate whether each material, system or method listed is relevant to your study. If you are not sure if a list item applies to your research, read the appropriate section before selecting a response.

## Materials & experimental systems

| n/a | Involved in the study |
|-----|----------------------|
| ☒ | Antibodies |
| ☒ | Eukaryotic cell lines |
| ☒ | Palaeontology and archaeology |
| ☐ ☒ | Animals and other organisms |
| ☒ | Clinical data |
| ☒ | Dual use research of concern |

## Methods

| n/a | Involved in the study |
|-----|----------------------|
| ☒ | ChIP-seq |
| ☒ | Flow cytometry |
| ☒ | MRI-based neuroimaging |

## Animals and other research organisms

Policy information about studies involving animals; ARRIVE guidelines recommended for reporting animal research, and Sex and Gender in Research

| | |
|---|---|
| Laboratory animals | Animals used:<br>C57BL/6 (3 females, 35 males; 2-16 months old)<br>APPnl-g-f heterozygotes (3 males, 4 females; 2-3 months old)<br>TH::Cre heterozygotes (line Fl12, www.gensat.org) (1 male; 16 months old)<br>Drd2::Cre heterozygotes (line ER44, www.gensat.org) (1 male, 2 females; 4 months old)<br>All animal procedures complied with relevant ethical regulations and were performed after approval by the Institutional Animal Care and Use Committee (IACUC) of Cornell University (protocol number 2015-0029). All mice were housed in a climate-controlled facility kept at 22o C and 40-50% humidity, under a 12-hour light-dark cycle with ad libitum access to food and water. |
| Wild animals | Study did not involve wild animals |
| Reporting on sex | For mouse inter-eye distance measurement, as this was a novel dataset, both male and female mice were used and data is reported disaggregated for sex. For all other experiments, only male mice were used to reduce variance in measurement and because these experiments replicated or extended results from prior publications where data on male mice were reported. |
| Field-collected samples | Study did not involve samples collected from the field |
| Ethics oversight | Ethics oversight provided by the Institutional Animal Care and Use Committee (IACUC) of Cornell University (protocol number 2015-0029). |

Note that full information on the approval of the study protocol must also be provided in the manuscript.

