## [Peer Review File · Nature Methods]

MouseGoggles: an immersive virtual reality headset for mouse neuroscience and behavior

Corresponding Author: Dr Matthew Isaacson

Version 0:

Decision Letter:

22nd Nov 2023

Dear Dr Isaacson,

Thank you for your patience. Your Brief Communication, "MouseGoggles: an immersive virtual reality headset for mouse neuroscience and behavior", has now been seen by three reviewers. As you will see from their comments below, although the reviewers find your work of considerable potential interest, they have raised a number of concerns. We are interested in the possibility of publishing your paper in Nature Methods, but would like to consider your response to these concerns before we reach a final decision on publication.

We therefore invite you to revise your manuscript to address these concerns. Specifically, please be sure to demonstrate the advantages over standard VR systems more thoroughly and discuss the limitations of your system. It won't be necessary to conduct experiments with a large number of animals.

Importantly, please discuss the similarities and differences of your system with that described in <https://www.researchsquare.com/article/rs-3352160/v1> and <https://patents.justia.com/patent/20220295743>. You will need to make a strong case for the uniqueness of your system.

Link Redacted

Note: This URL links to your confidential home page and associated information about manuscripts you may have

submitted, or that you are reviewing for us. If you wish to forward this email to co-authors, please delete the link to your homepage.

We hope to receive your revised paper within 2-3 months. If you cannot send it within this time, please let us know. In this event, we will still be happy to reconsider your paper at a later date so long as nothing similar has been accepted for publication at Nature Methods or published elsewhere.

OPEN SCIENCE REQUIREMENTS

REPORTING SUMMARY AND EDITORIAL POLICY CHECKLISTS

DATA AVAILABILITY

All novel DNA and RNA sequencing data, protein sequences, genetic polymorphisms, linked genotype and phenotype data, gene expression data, macromolecular structures, and proteomics data must be deposited in a publicly accessible database, and accession codes and associated hyperlinks must be provided in the "Data Availability" section.

CODE AVAILABILITY

Please include a "Code Availability" subsection in the Online Methods which details how your custom code is made available. Only in rare cases (where code is not central to the main conclusions of the paper) is the statement "available

upon request" allowed (and reasons should be specified).

For more information on our code sharing policy and requirements, please see:
<https://www.nature.com/nature-research/editorial-policies/reporting-standards#availability-of-computer-code>

MATERIALS AVAILABILITY

ORCID

Nature Methods is committed to improving transparency in authorship. As part of our efforts in this direction, we are now requesting that all authors identified as 'corresponding author' on published papers create and link their Open Researcher and Contributor Identifier (ORCID) with their account on the Manuscript Tracking System (MTS), prior to acceptance. This applies to primary research papers only. ORCID helps the scientific community achieve unambiguous attribution of all scholarly contributions. You can create and link your ORCID from the home page of the MTS by clicking on 'Modify my Springer Nature account'. For more information please visit <http://www.springernature.com/orcid>.

Best regards,
Nina

Nina Vogt, PhD
Senior Editor
Nature Methods

Reviewers' Comments:

Reviewer #1:
Remarks to the Author:

In this manuscript, Isaacson and colleagues present a novel version of the VR system for head-fixed mice that features a miniature contraption (hence entitled MouseGoggles), easy and cheap design, direct stimulation of the eyes with minimal stray light, and independent control of each eye stimulus. MouseGoggles VR functionality is illustrated using 2PM visual cortex and hippocampal electrophysiology experiments, as well as spatial reward association and innate looming fear paradigms. This work is certainly going to lower the bar for the use of VR in many labs with limited budgets due to the price, eliminate shortcomings of the conventional HD VR, as well as will enable VR experiments that are currently not possible. Hardware is built on inexpensive and widely available components, both optical, mechanical, and electronic designs and software are made open source. Experiments are performed at a high level of quality and the data and results are convincing. I have a few comments listed below. While the new form factor, low cost, stereo-ability, and compactness make this system clearly novel and show great promise, the present form of the manuscript doesn't demonstrate or elaborate on any of these advantages.

The minimalistic and low-cost hardware/software used in the MouseGoggles clearly leads to some compromises on the flexibility and performance of the VR implementation. Maximal frame rate, latency, and richness of the 3D cues will surely be limited compared to setups that rely on high-speed and high-resolution projector(s), GPU cards, and performant VR engines (e.g. Unity). It would be good if the authors laid out the cons of such a system for animal experiments and possible future directions for circumventing them.

A natural and important advance in the setup is the possibility to independently render images to both eyes thus enabling stereo-VR. However, unfortunately, the authors do not include such experiments. While testing V1 and CA1 cell tuning simply confirms that MouseGoggles is performing comparably to the existing conventional HF VR setups, these experiments

do not demonstrate the advantages of the new system and do not make the case why it should be published in this journal as a major advance of the technology. Some experiments contrasting mono vs stereo-VR would demonstrate this key advance in the technology that is missing in the neuroscience toolbox.

Innate behavior, such as startle response in response to looming stimulus, even though possibly recapitulating natural avoidance behavior observed in freely-moving rodents, is hardly making a decisive showcase for the system distinguishing it from the conventional ones. It is not clear from the methods section describing conventional projector VR that authors compare to if that system has used all possible means to increase projection surfaces at the lower visual field, including the floor (e.g. as done by labs of Mehta and Tank) by adding a cutout floor disk all around the ball. As many groups implemented this innate fear response with a simple overhead screen or sound, this paper doesn't put forward a distinct use-case for studying flight response to the looming stimulus under head-fixed conditions that could not be done in freely-moving animals.

Stray light measurements make the case for the clear improvement in that parameter over plain-vanilla imaging setup in which the optical axis is not shielded from the light. However, similar to the enclosure protecting the 2PM from the monitor light, it is possible (and done by some scientists) to design and 3D print an enclosure protecting the cranial window from the external stray light. The use of blue light is another option (e.g. Kuznetsova et al JNM 2021), which, I assume, authors refer to as "filtering". Besides, a significant amount of stray light reaches the brain through the eye, which would mean that if imaging was performed in more frontal regions, high-intensity eye stimulation might cause more stray light diverted through this pathway than in a conventional setup. Could authors measure that? What is the functional consequence of the difference in the stray light for the performance of 2PM imaging? Arguably somatic high-SNR Ca imaging will likely not be affected. I suggest that to make this feature a decisive one that sets this setup apart from the conventional one, there needs to be a clear demonstration of the functional advantages.

What appears to be the top point setting the system apart is that, in addition to enabling stereo VR, it is inexpensive and compact. Inexpensive - means that scalable, high-throughput parallel VR setups would allow training the large batches of mice, something that, clearly was not possible so far. However obvious, this feature is not emphasized in the manuscript. Compactness of the setup is only slightly outlined by the authors, but not exploited. Rotations of the miniaturized setup allow easy implementation of the proper 2D VR (similar to Chen et al eLife 2018). Rotations around other than the vertical axis allow for so far impossible experiments testing vestibular input contributions. Realistic vestibular inputs could be added by implementing appropriate movements of the apparatus using hexapods.

Reviewer #2:

Remarks to the Author:

Mouse Goggles is a novel VR display device for the binocular visual stimulation of mice. Together with its online documentation this is to be a fully documented open-source tool that others can replicate. The results of MouseGoggle experimental testing confirm that it functions at least as well as traditionally used surround screens with visual stimuli or VR scenes. The experimental testing for MouseGoggles has been done comprehensively. The report is brief, to the point, well written, and complete.

The usefulness of the method will depend on the possibility that it can actually be used as an open source tool and replicated by others.

Authors should discuss how their relatively simple optical solution could potentially be improved. In this context they should cite the publicly accessible manuscript about Mocolus doi.org/10.21203/rs.3.rs-3352160/v1.

Authors should discuss if patent PCT HU2020/050029 published under WO2021009526A1 covers or is different from their method as it may affect its use as an open source tool (accessible e.g. at <https://patents.justia.com/patent/20220295743>).

The authors should improve the online documentation that is cited in the manuscript in order to better facilitate replication of this open source method.

Mechanics, CAD:

1. Please upload not only the STL files but also the STP files or IPT drawing of the CAD designs for easier modification
2. The bill of materials mentions all parts in [cm] however Figure 2 shows them in [mm] – please update the bill of materials and use only one unit of measurement to avoid confusion
3. Recommendation for a future revision: Reconsider the material that was used in the printing. If the parts will be adjustable or moveable at some point, resin material will become brittle quickly.

Electronics:

1. Upload not only gerber files but also the schematic and PCB file for easier modification
2. Include the fabrication specifications for the PCBs: board thickness, copper thickness and surface finish
3. In MinibrdV1, the copper to copper spacing is only 0.06 mm – for a future revision I recommend updating the layout to increase the minimum copper to copper spacing to at least 0.2 mm; this simplifies demands on production.
4. In MinihatV1, there are 90 degree traces and these will form acid traps that cause corrosion – for a future revision I recommend updating the layout with traces only at 45 degrees

Software:

- a. The driver for sending the images via SPI to two small screens is from an external project (fbcpl-ili9341 and fbcpl-st7789) which seems to be well documented, no action required.

b. Drivers for detecting the movement of the ball.

The movement is transmitted to the PC/RasPi as if it were a PC mouse. The ADNS3080 sensor is used.

"mouseVRheadset_controller_V4.ino" is not commented in detail. This is so small that this is not of great importance. Action: add a few explanatory sentences to facilitate faster orientation.

The purpose and use of the files in the "ADNS3080 debugging" folder must be explained. While these are only for debugging, how and what they are used for is not clear. This is also because these data have no comments. Action required: Explain and comment.

c. The experiments with 3D glasses use the Godot engine. Some examples are provided. But you need some knowledge about this engine, especially because the experiments run completely within the engine. This engine is not only responsible for the representation of the 3D world, but also for the process and the logging of the data. The examples make it easier to get started, but still require some familiarisation with Godot. Recommended action: the authors might think about how they could further facilitate getting familiar with this part of the code.

d. The glasses are also available in a variant for "Monocular Display". This uses the Teensy board to control a small display. "GC9307_teensy_GFX.ino" has a lot of comments on single lines, but a more general documentation to understand the context of the parameters would be nice. Especially because there are so many parameters. Action required: provide a better general documentation and include a table of all parameters and their use.

While the software is neither very large nor a complex project, with its current level of documentation it still poses a challenge to find ones way around. This does not apply to the external SPI monitor driver. This part alone is much larger and more complex than MouseGoggles.

Minor comments

551 check sentence for completeness: ... last bin excluded due the mouse's constrained position ...

555 For ease of understanding change: "All licks not occurring following a reward delivery were defined as "exploratory licks". TO Licks at other times than after a reward delivery

619 a 2nd cohort of mice were tested -> was tested

Reviewer #3:

Remarks to the Author:

Virtual reality has become a powerful tool for studying a variety of brain functions in head-fixed rodents. Current implementations of rodent VR involve large displays that not only create difficulties for the experimenter, but also do not provide an immersive experience for the animal, largely due to their expansive visual field. Isaacson, Chang, et al. designed a head-set based VR system to overcome current constraints. They provide evidence that their system solves many of the practical issues with VR, including reduced light pollution for imaging methods and reduced cost. They use neural recordings to show that the visual tuning properties of neurons are comparable to results obtained with standard techniques, supporting the quality of image presentation. Finally, they provide evidence that the VR experience for the mouse is immersive through effective behavioral training in reward and innate fear behaviors.

This work represents a very useful advance for both the quality of rodent VR experience and the ability of individual laboratories to implement VR systems, especially given the low-cost and open source design. Overall, the authors provide a straightforward and convincing case for the improvements this system makes over traditional rodent VR systems. The manuscript could be substantially improved by addressing the following points:

-The system provides 230 deg of field coverage in the horizontal azimuth, with 140 deg per eye and 25 deg of binocular overlap. The authors should report measured values of the mouse visual field from previous studies in the main text, and discuss the limitations of the system in obtaining full coverage in the Discussion. Also, the approximate elevation covered by the system should be reported in the main text.

-How does the 130 ms input to display latency compare to other rodent VR systems? My understanding is that latencies greater than 20ms are detectable to humans, so it seems possible this relatively long latency may limit the immersive experience for the mouse. The authors should discuss this.

-The measurements in 1b-d are in the range of 0-70 deg, but the authors report monocular coverage of 130 deg. Is there some reason why they did not model the full coverage, which is nearly twice the range in the figure?

-The modeling of the light reaching the eye from the screen/lens is very useful. It would also be useful to have a measurement of the performance of the actual system, e.g., by projecting onto a model or real mouse eye.

-The comparison of light pollution between the monitor and monocular display is quite striking. It would be ideal to see a similar direct comparison of the two systems for the visual tuning properties in Fig2d-i, especially since the stimulus used for receptive field mapping is different from the referenced study (Niell & Stryker, 2008).

-How do the tuning properties of CA1 place cells compare to other mouse VR studies and freely moving mouse studies (e.g. place field width)?

-In comparing the behavioral responses to looming stimuli on MouseGoggles vs. a traditional projector system, was the

projector system oriented above the mouse? Based on the methods, it appears the projector reached a maximum of only 64 deg elevation, so is this the presumed reason that a response was not elicited? The ability to adjust elevation with MouseGoggles is indeed useful, but ideally the behavior would be compared to a projector system with similar visual field coverage. If it is experimentally infeasible to adjust the projector system to cover a comparable region of the visual field, the authors should state this. Otherwise, the experiment should be conducted with comparable overhead coverage between the two systems.

-It appears that video of the mouse was acquired during the looming stimulus presentation (Fig3f and methods). The methods clearly state the responses were manually scored, but the main text and figure do not mention this, and the line fit to the mouse's back in Fig3f might lead the reader to assume the quantification of behavior was achieved with computational video analysis. Is the resolution of the video sufficient to perform the startle analysis? If so, the authors should quantify the behavior using image analysis methods or markerless pose estimation. If not, the authors should clearly state that in the text/Fig3g that the quantification was performed by two human observers. The authors could also plot the velocity of the animal measured via the spherical treadmill aligned to the onset of the looming stimulus.

-Does this system permit the measurement of eye movements? Given that mice move their eyes (though somewhat infrequently) while head-fixed, it would be ideal to measure these movements during stimulus presentation. The authors mention that pupil tracking would be feasible with this system; how would this be achieved?

-Face/whisker movements contribute significantly to neural activity. Does this system allow for these measurements, or do the MouseGoggles fully obstruct the view?

-If possible, the experimenters should quantify whisker contact with the system. Do any of the whiskers touch the system, and if so with what frequency? Extensive whisker contact with the system would presumably degrade any immersive experience the mouse may have, and would make it difficult to perform some of the multisensory experiments the authors propose in the Discussion.

-How do the authors calibrate the system with each individual mouse, given e.g., the inherent variability in headplate placement?

-Given that monocular cues should be sufficient for most experiments performed in the manuscript (formation of place cells, responses to looming stimuli, reinforcement learning), an "immersive" experience may not be requisite for the results obtained here. A more convincing test that the animal is having an immersive experience would be ideal, e.g., behavior with a virtual cliff showing that animals stop to avoid falling over the edge.

Version 1:

Decision Letter:

Our ref: NMETH-A53654A

31st Jul 2024

Dear Dr. Isaacson,

Thank you for your patience during this period and for submitting your revised manuscript "MouseGoggles: an immersive virtual reality headset for mouse neuroscience and behavior" (NMETH-A53654A). It has been seen by the original referees and their comments are below. The reviewers find that the paper has improved in revision, and therefore we'll be happy in principle to publish it in Nature Methods, pending minor revisions to satisfy the referees' final requests and to comply with our editorial and formatting guidelines.

TRANSPARENT PEER REVIEW

Please note: we allow redactions to authors' rebuttal and reviewer comments in the interest of confidentiality. If you are concerned about the release of confidential data, please let us know specifically what information you would like to have removed. Please note that we cannot incorporate redactions for any other reasons. Reviewer names will be published in the

peer review files if the reviewer signed the comments to authors, or if reviewers explicitly agree to release their name. For more information, please refer to our [FAQ page](https://www.nature.com/documents/nr-transparent-peer-review.pdf).

ORCID

IMPORTANT: Non-corresponding authors do not have to link their ORCID but are encouraged to do so. Please note that it will not be possible to add/modify ORCID at proof. Thus, please let your co-authors know that if they wish to have their ORCID added to the paper they must follow the procedure described in the following link prior to acceptance: <https://www.springernature.com/gp/researchers/orcid/orcid-for-nature-research>

Best regards,
Nina

Nina Vogt, PhD
Senior Editor
Nature Methods

Reviewer #1 (Remarks to the Author):

Authors addressed all my comments, I would fully support publishing this paper.

Reviewer #1 (Remarks on code availability):

Software and hardware are well-documented and all open-source.

Reviewer #2 (Remarks to the Author):

The authors have made quite some effort to respond to my concerns.

Comment 14

The "Hardware" directory has been renamed to "Other Hardware". This was a good decision, as it helps to differentiate between MouseGoggles and other hardware mentioned in paper. The added documentation should now enable and simplify the initial setup of treadmill and the sensors.

Comment 15

Still no comments in the code. But now it is explained how these files should be used and their purpose. Since the files are small, this seems fine now.

Comment 16

The new short "Introduction to Godot" should indeed make it a little easier to get started. Even if there is still a relatively large hurdle here, as Godot has to be learned. But the experiments seem to be both very small and very similar, so that customisation should be possible.

Small note and suggestion for a further small change: it should be noted which version of Godot is used here. I opened the project with 4.2.2 and had some error messages in the script editor. This happened although the documentation explicitly calls for the last version of Godot. However, this probably refers to the latest 3.x version.

Comment 17

The description of the hardware and the hardware assembly instructions should now be sufficient to build the system. And the software description/documentation should help to setup the system.

Comment 18

I agree.

Reviewer #3 (Remarks to the Author):

The updated manuscript provided by Isaacson, Chang et al. entitled "MouseGoggles: an immersive virtual reality headset for mouse neuroscience and behavior" represents a substantial improvement over the initial manuscript submission. The authors responded to the reviewer comments with a significant amount of work that addresses the issues with the original manuscript, including:

- Relationship of the system to the total visual field
- Discussion of latency and steps toward reducing it
- Measured the light reaching the eye
- Clarification on visual stimuli

Quantification of place cell properties
Improved behavioral measures during looming behavior
Characterization of whisker obstruction

The heroic addition of pupillometry also sets this system apart from other emerging immersive mouse VR systems, making this system particularly useful to neuroscientists who also want to track pupil position and size.

I have only a minor comment, which is that the authors should consider mentioning a finding from Meyer et al., Curr Bio 2020, that might explain the eye movements in response to the looming stimulus. Eye movements in headfixed mice appear to result primarily from attempted head movements (see Fig S5 in that paper), so one interpretation of the results here is that the animals are attempting to escape by moving their heads, which results in a saccade. While a subtle point, the authors' interpretation that the animal's gaze is being directed toward the stimulus is more likely due to an artifact of the animal's inability to move its head freely.

Dear Dr. Vogt,

We thank you for the opportunity to submit our revised manuscript, "MouseGoggles: an immersive virtual reality headset for mouse neuroscience and behavior", to Nature Methods. We appreciate the time taken by you and the reviewers to comment and make suggestions on both the technical and experimental aspects of our approach, and in responding to these comments we have made substantial improvements to our manuscript. These improvements include a more comprehensive comparison of our MouseGoggles system to existing VR technologies and recently developed headset-based systems, an improved online repository to enhance the reproducibility and modifiability of our technology, and the addition of a substantial new capability to our VR system – binocular eye and pupil tracking during VR – which has been our most requested new feature and sets MouseGoggles apart from other recently reported headset-based mouse VR systems. These improvements are demonstrated with the addition of 13 main figure panels, 14 supplemental figure panels, 2 supplementary videos, new results, discussion, and methods text, and substantial additions to the public online repository. These changes are outlined in our point-by-point response to all comments below.

Comments from the Editor:

Comment 1: *Specifically, please be sure to demonstrate the advantages over standard VR systems more thoroughly and discuss the limitations of your system.*

Response: Thank you very much for this suggestion. We have added new details and results to the main text and methods sections expanding on the advantages of our system over traditional panoramic VR systems. These advantages include a substantially reduced form factor and lowered computational requirements (enabling low-cost VR in a small footprint and a greater ability to scale up experiments, detailed on lines 167-169), increased immersivity and ability to study innate behaviors during head-fixation (lines 148-152), and the ability to perform binocular eye and pupil tracking during VR (lines 127-142, 169-170, and demonstrated in Fig. 4 and Supplementary Video 3). We have also added some discussion of the limitations of our system, including the relatively low resolution displays used which are well-suited for mouse visual acuity but could be difficult to be directly used on other animals models with higher visual acuity, the 130 ms input-to-display latency which may be problematic during closed-loop feedback of fast behaviors or neural events, and the occlusion of many of the mouse's whiskers by the headset which may confound the sensory experience of a mouse navigating virtual environments (lines 154-165). We also discuss potential future hardware and software improvements that could increase display resolution and lower latency (lines 158-165), while a new supplemental figure characterizes the amount of whisker contact at different headset positions to provide guidance on minimizing this potential sensory conflict (Supplementary Fig. 7).

Comment 2: *Importantly, please discuss the similarities and differences of your system with that described in <https://www.researchsquare.com/article/rs-3352160/v1> and <https://patents.justia.com/patent/20220295743>. You will need to make a strong case for the uniqueness of your system.*

Response: Since our submission of this manuscript in Aug. 2023, two new mouse VR headset systems have been described in the literature. The 1st system, named "Moculus," was released as a preprint on Research Square in Oct. 2023 (<https://www.researchsquare.com/article/rs-3352160/v1>) and cited patent applications from previous years. The 2nd system, named "iMRSIV," was published in Dec. 2023 in Neuron (<https://www.cell.com/neuron/fulltext/S0896->

6273(23)00893-0) and built using the same software platform as Moccus but using a different display and lens configuration to support a larger field of view. In our manuscript, we now compare our MouseGoggles system with these two new mouse VR headsets. There are several similarities: small form factor, greater accessibility and scalability of head-fixed mouse VR, and evidence of experimental benefits from increased mouse immersion in the virtual world with headset-based systems (lines 148-152). We have also added new discussion and results on the uniqueness of our system compared to these other approaches, including the ability to perform binocular eye and pupil tracking during VR, the open hardware and software platforms used by MouseGoggles (Raspberry Pi and Godot game engine; all details for replication of hardware and software on GitHub), reduced cost and computational requirements, simplified optical design requiring no custom lenses or precise eye alignment protocols, and the greater mobility of the miniature headset to accommodate rotating head mounts. These discussion points are detailed in the main text (lines 154-170), with headset mobility and rotation demonstrated in Supplementary Video 4 and binocular eye and pupil tracking demonstrated in Fig. 4 and Supplementary Video 3. Taken together, our and other headset-based VR systems have shown increased immersion, as well as reduced form factor and cost relative to panoramic systems. These benefits will be salient for nearly all neuroscience researchers using VR systems, and rapid adoption is likely.

Our MouseGoggles system is distinct from the other two systems demonstrated so far primarily in having the ability to track eye gaze direction (to assess what part of the VR space the mouse is looking at) and pupil diameter (a reliable indicator of arousal), with many additional minor differences. This eye tracking capability has been, by far, the most requested addition to MouseGoggles that we have heard from other researchers when they learn about our system. We are happy that we were challenged to make this addition to the system now by both a Reviewer and the need to make a strong case for the uniqueness of our system.

Comments from Reviewer 1:

Comment 1: In this manuscript, Isaacson and colleagues present a novel version of the VR system for head-fixed mice that features a miniature contraption (hence entitled MouseGoggles), easy and cheap design, direct stimulation of the eyes with minimal stray light, and independent control of each eye stimulus. MouseGoggles VR functionality is illustrated using 2PM visual cortex and hippocampal electrophysiology experiments, as well as spatial reward association and innate looming fear paradigms. This work is certainly going to lower the bar for the use of VR in many labs with limited budgets due to the price, eliminate shortcomings of the conventional HD VR, as well as will enable VR experiments that are currently not possible. Hardware is built on inexpensive and widely available components, both optical, mechanical, and electronic designs and software are made open source. Experiments are performed at a high level of quality and the data and results are convincing.

Response: We thank the reviewer for this positive assessment of our open-source system's advancements over traditional VR technology (e.g. increased accessibility, reduced technical shortcomings, and utility in enabling new VR experiments) and in the high quality of data demonstrating its use.

Comment 2: I have a few comments listed below. While the new form factor, low cost, stereo-ability, and compactness make this system clearly novel and show great promise, the present form of the manuscript doesn't demonstrate or elaborate on any of these advantages.

Response: As detailed in responses to the specific comments below, we have added additional results and discussion points that explicitly describe these advantages of the system. This includes discussion of the ability to investigate binocular integration and stereo vision with

independent eye stimulus control (response to comment 4), to scale up VR experiments due to our system's low cost and small size (response to comment 8), and to rotate the VR system around or with the animal to engage the animal's vestibular system (response to comment 9).

Comment 3: The minimalistic and low-cost hardware/software used in the MouseGoggles clearly leads to some compromises on the flexibility and performance of the VR implementation. Maximal frame rate, latency, and richness of the 3D cues will surely be limited compared to setups that rely on high-speed and high-resolution projector(s), GPU cards, and performant VR engines (e.g Unity). It would be good if the authors laid out the cons of such a system for animal experiments and possible future directions for circumventing them.

Response: We thank the reviewer for this comment and apologize for not discussing these limitations in more depth in the original manuscript. The primary technical compromise of MouseGoggles relative to other systems (including human VR systems, traditional panoramic mouse VR systems, and the two other headset-based mouse VR systems) is in display resolution, though we are confident this does not hinder mouse visual neuroscience applications. We used small 240x210 or 240x240 pixel circular displays that contain approximately 1/50th the number of pixels as a single 1080p HD monitor, which substantially reduces the smallest simulated objects and features that can be resolved on screen. However, as prior literature suggests and our experiments have demonstrated (Fig. 2), this reduction in resolution is not a detriment to stimulating the mouse visual system due to the mouse's relatively poor visual acuity, being approximately 1/100th that of humans (~60 cpd maximum for humans vs ~0.5 cpd for mice, reported in Sinex et al, Vision Res. 1979). To better demonstrate this poor visual acuity of the mouse visual system, inserted below is an example image demonstrating how a high-resolution image (left) and a low resolution MouseGoggles image (right) might appear to the mouse (bottom row):

Low resolution displays would be a limitation if MouseGoggles were to be adapted for a different animal model with higher visual acuity (rats or tree shrews, for example – ongoing in our lab), and would need to be circumvented with the use of higher-resolution displays, and potentially heftier computational resources. On the other hand, because we do not need higher resolution due to the poor visual acuity of mice, an added benefit is a substantial reduction in the computer processing power and data transfer rates required to render 3D scenes and stream images to the displays. With the display resolution we use, complex 3D environments can be rendered at high frame rate on a Raspberry Pi 4, which only has a simple onboard GPU, and frames can be streamed at up to 80 fps using a simple SPI interface. To achieve a similar result, other VR systems require an expensive desktop computer and powerful GPU (e.g. Nvidia RTX3070 used by iMRSIV). Our low-cost solution makes VR research more accessible and promotes high-throughput VR experiments.

As the reviewer noted, many VR systems used in neuroscience use the Unity game engine to generate 3D scenes and create experiments. Compared to very high-performance game engines (e.g. Unreal Engine), Unity is a beginner-friendly engine more popular among researchers. For MouseGoggles, we opted for the game engine “Godot”, in part due to the ethos of our project relying on open-source tools (unlike Unity, Godot is a fully free and open-source engine). Additionally, despite being lightweight and beginner-friendly, Godot is a full-featured 3D game engine, capable of complex lighting, shaders, particles, and physics simulations, which we are confident is more than powerful enough to support all mouse neuroscience use cases. Researchers may find Godot even easier to work with than Unity as it uses a node-based graphical user interface to create 3D scenes and uses Python-like scripts for game design, instead of Unity’s C# programming. To assist researchers in developing with the

Godot game engine, we have added an “Introduction to Godot” document along with a Godot project file containing seven example VR experiments in the online repository

Notes on the motivation to use the Godot game engine and on VR environment development have been added to the main text and methods (lines 165-167, 399-403).

Comment 4: A natural and important advance in the setup is the possibility to independently render images to both eyes thus enabling stereo-VR. However, unfortunately, the authors do not include such experiments. While testing V1 and CA1 cell tuning simply confirms that MouseGoggles is performing comparably to the existing conventional HF VR setups, these experiments do not demonstrate the advantages of the new system and do not make the case why it should be published in this journal as a major advance of the technology. Some experiments contrasting mono vs stereo-VR would demonstrate this key advance in the technology that is missing in the neuroscience toolbox.

Response: The reviewer is correct in that we have not demonstrated an experiment that utilizes stereo VR over mono VR, although all VR scenes were rendered stereoscopically. We have put considerable thought into the possibility of a convincing mono vs stereo VR demonstration experiment, but our reading of the literature on stereoscopic vision in mice suggests that a meaningful stereo VR experiment is beyond the scope of this manuscript due to ongoing uncertainty about the role of stereoscopy in natural mouse behavior.

Some limited aspects of stereo vision have been previously reported in mice using custom stereo displays: visual cortical neurons show some stereo-disparity selectivity (Chioma et al, J Neurosci 2020: <https://pubmed.ncbi.nlm.nih.gov/33051348/>), mice can be trained to use this information to discriminate “near” and “far” surfaces in random dot stereograms (Samonds et al, J Neurosci 2019: <https://www.ncbi.nlm.nih.gov/pmc/articles/PMC6786824/>), and there has even been a report describing the innate use of stereo cues in a novel angled pole descent test (Boone et al, Cur Bio 2021: [https://www.cell.com/current-biology/fulltext/S0960-9822\(21\)00272-4](https://www.cell.com/current-biology/fulltext/S0960-9822(21)00272-4)). In sum, however, these reports describe rather limited mouse stereo vision capabilities: mice don’t use vergence eye movements to fixate by depth like primates do, and the neural selectivity to stereo disparity was relatively weak and only relevant over a narrow range of disparities. Other animal models such as tree shrews show comparably more selective responses to stereo disparity (Tanabe et al, Curr Bio 2022: <https://pubmed.ncbi.nlm.nih.gov/36417902/>) and may be better models to study stereo vision (this has motivated us to begin development of a larger, higher-resolution version of MouseGoggles suitable for rats and tree shrews). Nonetheless, stereoscopy is a fascinating topic in mouse neuroscience, and MouseGoggles could be helpful in future VR experiments that explore stereoscopic vision in mice. For example, one could develop a stereo vs. mono VR version of the angled pole descent test, which is necessary over a standard flat visual cliff as the downward angle of the descent shifts the view of the cliff into the binocular overlap region where stereo vision might contribute to depth perception.

Because prior studies have already described custom stereo displays to study stereoscopy in mice, our manuscript focuses on more novel aspects of our method, such as the greater immersivity as well as eye and pupil tracking in VR. But we fully agree that the stereoscopic capabilities of MouseGoggles is another important advance of our method that should be valuable to the neuroscience community, so we have highlighted this feature in the discussion (line 149).

Comment 5: Innate behavior, such as startle response in response to looming stimulus, even though possibly recapitulating natural avoidance behavior observed in freely-moving rodents, is hardly making a decisive showcase for the system distinguishing it from the conventional ones. It is not clear from the methods section describing conventional projector VR that authors compare to if that system has used all possible means to increase projection surfaces at the

lower visual field, including the floor (e.g. as done by labs of Mehta and Tank) by adding a cutout floor disk all around the ball. As many groups implemented this innate fear response with a simple overhead screen or sound, this paper doesn't put forward a distinct use-case for studying flight response to the looming stimulus under head-fixed conditions that could not be done in freely-moving animals.

Response: The reviewer is correct that innate fear reactions to looming visual stimuli have been observed using VR setups in freely-moving rodents. However, the primary goal of this experiment in our manuscript was to determine if looming reactions could also be elicited in head-fixed conditions, which is necessary for many setups that use neural recording systems not amenable to free-walking. The panoramic VR system we used is state-of-the-art and covers a very wide field-of-view, leaving only a small cutout above the mouse to accommodate a microscope objective and a larger gap below the mouse to fit a spherical treadmill. This projector setup did not include the cutout floor disk approach the reviewer describes, though because the looming object approached at an angle from overhead and did not produce a ground shadow, this may not be a meaningful difference. As reviewer 3 (in comment 8) pointed out, the projector also did not have quite the same vertical extent as MouseGoggles, though in new looming stimulus experiments with eye and pupil tracking (Fig. 4), we blacked out the top of the MouseGoggles displays to match the projector's vertical extent and startle reactions were still observed, suggesting that the extent of visual field coverage does not explain the difference. To our knowledge, only our manuscript and the recent publication on the iMRSIV headset system report fear-based reactions to looming stimuli in head-fixed mice, and neither could replicate this startle reaction in a panoramic VR setup. Thus, we believe that the prior and new data on mouse responses to looming stimuli with headset VR does demonstrate increased immersion in the VR environment, which could be broadly helpful for VR-based experiments.

Beyond head-fixed neural recording, there are other benefits to being able to perform this task during head-fixation, such being able to use precise behavioral monitoring systems that are more effective in stationary animals. In our revision we have added a new demonstration of one such system: binocular eye and pupil tracking. By building eye tracking cameras into the VR headset, we monitored eye and pupil dynamics in both eyes simultaneously during the presentation of looming stimuli. We observed consistent eye tracking in the direction of the overhead approaching virtual object, and an increase in pupil diameter following the looming stimulus that diminishes with further repetitions of the stimulus, similar to the diminishing startle response we reported in the original manuscript. The ability to elicit fear responses in head-fixed setups where not only neural activity but also eye and pupil dynamics can be recorded, thus enabling assessment of the mouse's attention, arousal, and memory encoding is a substantial advance for mouse neuroscience, in particular the study of emotional circuits. This new MouseGoggles design with eye tracking and application to the head-fixed looming experiment is described in the main text (lines 127-142) and shown in Fig. 4 and Supplementary Video 3.

Comment 6: Stray light measurements make the case for the clear improvement in that parameter over plain-vanilla imaging setup in which the optical axis is not shielded from the light. However, similar to the enclosure protecting the 2PM from the monitor light, it is possible (and done by some scientists) to design and 3D print an enclosure protecting the cranial window from the external stray light. The use of blue light is another option (e.g. Kuznetsova et al JNM 2021), which, I assume, authors refer to as "filtering".

Response: On the issue of stray light contamination into sensitive imaging applications, the reviewer accurately points out that shielding the cranial window (or what we described as "shielding the objective" in the manuscript) or filtering out display light (e.g. using blue light with a short-pass color filter, since blue LEDs used in commercial displays typically emit small amounts of longer wavelength light) can be effective in enabling imaging during high intensity visual stimulation. The main advantage of using our MouseGoggles display is that this additional

shielding or filtering is not necessary. This point could range from being irrelevant for setups that already have appropriate shielding solutions in place, to a significant advantage for setups that would otherwise have to create and test new strategies appropriate for their equipment and experiment. Some imaging setups feature components that make shielding and filtering more difficult, such as those that use multiple different microscope objectives or use blue/green fluorescent probes, so having a visual stimulation system which produces minimal stray light to begin with can be a benefit. Because our display system was designed with the goal that it could simply be “dropped in” to most existing experimental setups, we believe that this issue of stray light reduction is important in that context, though we agree with the reviewer’s point that this is otherwise a surmountable problem. We have adjusted the main text so that this feature is described as an improvement over a standard off-the-shelf monitor, and equivalent to a monitor with substantial added shielding (lines 82-85, see also our response to the next comment). New data on the comparison to a shielded monitor is demonstrated in Supplementary Fig. 4 and described in the methods section (line 573-578).

Comment 7: Besides, a significant amount of stray light reaches the brain through the eye, which would mean that if imaging was performed in more frontal regions, high-intensity eye stimulation might cause more stray light diverted through this pathway than in a conventional setup. Could authors measure that? What is the functional consequence of the difference in the stray light for the performance of 2PM imaging? Arguably somatic high-SNR Ca imaging will likely not be affected. I suggest that to make this feature a decisive one that sets this setup apart from the conventional one, there needs to be a clear demonstration of the functional advantages.

Response: The reviewer brings up an excellent point that a small amount of light from the displays may reach imaging cameras or PMTs by passing through the pupil and scattering through the brain, and while the overall amount of light generated by a MouseGoggles display is substantially less than that of a full-sized monitor or projector, it is not immediately clear whether MouseGoggles would focus more or less light through the eye than traditional displays.

To investigate this possibility, we recorded light levels from both a MouseGoggles monocular display and a traditional flat screen with and without shielding the cranial window. In anesthetized mice, we imaged in both visual area 1, where visual stimulus-evoked neural activity could be seen from GCaMP6f fluorescence, and in somatosensory area 1, where no stimulus-evoked GCaMP activity was seen and the contribution from stray light could be quantified. To determine whether this measured stray light was acquired through an external pathway or an internal one (i.e., through the pupil and scattering through the skull), we compared recorded light levels with the cranial window either unblocked, where light could be collected from both internal and external pathways, or blocked, where light is only collected from external pathways. We found that using MouseGoggles or a substantially shielded traditional monitor, we could eliminate nearly all of the external stray light, where only minimal external stray light was recorded at maximum display brightness, far less than the intensity of typical GCaMP6s-labeled cell baseline fluorescence. In addition, we did not detect any additional light from the display reaching our detectors when the cranial window was unblocked relative to the open cranial window condition, suggesting that the internal pathway is not contributing any detectable stray light in our imaging setup, at least in the brain regions we imaged. We thus conclude that light contamination through the eye is not a major issue with MouseGoggles, and that total stray light from MouseGoggles is comparable to a highly shielded traditional monitor. These new results have now been included in Supplementary Fig. 4 and referred to in the main text (lines 82-85).

Comment 8: What appears to be the top point setting the system apart is that, in addition to enabling stereo VR, it is inexpensive and compact. Inexpensive - means that scalable, high-

throughput parallel VR setups would allow training the large batches of mice, something that, clearly was not possible so far. However obvious, this feature is not emphasized in the manuscript.

Response: We thank the reviewer for this helpful perspective and suggestion. Since the MouseGoggles system is even more compact and inexpensive than other recent mouse VR headset systems (Moculus and iMRSIV) and has substantially reduced computational requirements (only needing a Raspberry Pi 4 with no GPU), our system is particularly well suited to the scaled-up VR training the reviewer contemplates. We have added an additional discussion point highlighting this benefit (lines 165-169). To further demonstrate the ability to scale up VR experiments with small-footprint VR systems, we have added a description of a VR setup using a linear treadmill (lines 651-658) that fits within a 14x14 cm footprint (Fig. 4b) and added new documentation to the online repository with assembly instructions and code for this setup. We have also included instructions in our online repository ([REDACTED]) for controlling the VR system with VNC, so that multiple VR setups can be controlled from a single host computer, which may be the desired solution for running VR experiments in parallel.

Comment 9: Compactness of the setup is only slightly outlined by the authors, but not exploited. Rotations of the miniaturized setup allow easy implementation of the proper 2D VR (similar to Chen et al eLife 2018). Rotations around other than the vertical axis allow for so far impossible experiments testing vestibular input contributions. Realistic vestibular inputs could be added by implementing appropriate movements of the apparatus using hexapods.

Response: We agree with the reviewer that the compact size of MouseGoggles could allow rotations of the headset during use. Rotating the VR setup to test vestibular inputs is a fascinating possibility, though beyond the scope of this manuscript to thoroughly implement. We have added new discussion points about this possibility in the main text (lines 172-173). Additionally, since relatively recent evidence has shown that rotatable head mounts improve 2D navigation in VR (such as in Chen et al, eLife 2018; this citation has been added to our manuscript), we believe this could be a very important feature for future MouseGoggles implementations. To demonstrate how headset rotations could be measured for closed-loop experiments, we have added a new supplementary video demonstrating headset rotation, using an integrated sensor (accelerometer and gyroscope) and magnetometer. The accelerometer provided feedback for headset roll and the magnetometer worked well for yaw feedback (Supplementary Video 4). We have added details on this modification to the methods section and online repository for building and controlling these setups. We hope this addition will enable interested researchers to take on the scientific questions the reviewer mentions.

Comments from Reviewer 2:

Comment 1: Mouse Goggles is a novel VR display device for the binocular visual stimulation of mice. Together with its online documentation this is to be a fully documented open-source tool that others can replicate. The results of MouseGoggle experimental testing confirm that it functions at least as well as traditionally used surround screens with visual stimuli or VR scenes. The experimental testing for MouseGoggles has been done comprehensively. The report is brief, to the point, well written, and complete.

Response: We thank the reviewer for this positive assessment of our work.

Comment 2: The usefulness of the method will depend on the possibility that it can actually be used as an open source tool and replicated by others.

Response: This is an excellent point, and we are in full agreement with the reviewer. One of our primary goals in this project is that not only is it possible for others to replicate this method, but that it be simple and easy for non-experts to do. To that end, we have opted for off-the-shelf parts wherever possible (e.g. no custom lenses), have created a comprehensive parts list, and provide simple assembly and installation instructions that have all necessary details for replication. We have also verified that this system has been successfully replicated by multiple external labs through the online resources, though we acknowledge that our open-source documentation at the time of submission still had significant room for improvement. As detailed in response to many comments below (comments 6-18), we have made significant improvements to the online repository to facilitate further replication and modification by the research community.

Comment 3: Authors should discuss how their relatively simple optical solution could potentially be improved. In this context they should cite the publicly accessible manuscript about Mocus doi.org/10.21203/rs.3.rs-3352160/v1.

Response: Since the submission of our MouseGoggles manuscript, there are now two other headset-based mouse VR systems in the literature: Mocus (Rozsa et al, Research Square 2023; preprint: <https://www.researchsquare.com/article/rs-3352160/v1>) and iMRSIV (Pinke et al, Neuron 2023; published: [https://www.cell.com/neuron/fulltext/S0896-6273\(23\)00893-0](https://www.cell.com/neuron/fulltext/S0896-6273(23)00893-0)). We have added new discussion points commenting on the different optical design strategies used and their pros/cons. Importantly, the optical design we chose is in service of a primary goal of our project: that MouseGoggles is simple and easy to replicate and use. By using just a single off-the-shelf Fresnel lens, no custom lenses have to be ordered or cut to size. Also, because of our use of a display system positioned at infinity focus from the perspective of the mouse, no complicated eye positioning protocols (such as those used by Mocus and iMRSIV) are needed. To add a new test and demonstration of the performance of our optical design, we imaged the display's projection on the back plane of an enucleated mouse eye, confirming that our design produces a clear image of the display that is robust to small changes in eye position within the allowed space of the eyepiece. These new results are shown in Supplementary Fig. 1 and referenced in the main text (lines 50-52) and described in the methods section (lines 425-439). The primary disadvantage of this simple optical design is that the field of view coverage of our eyepieces is limited to 140 deg, whereas more complex optical designs (e.g. iMRSIV) can support increased field of view coverage up to 180 deg. However, this enhanced field of view comes at the expense of necessitating more precise eye positioning. These pros and cons are now detailed in the discussion (lines 159-162).

Comment 4: Authors should discuss if patent PCT HU2020/050029 published under WO2021009526A1 covers or is different from their method as it may affect its use as an open source tool (accessible e.g. at <https://patents.justia.com/patent/20220295743>).

Response: As it currently stands, the Mocus patent application was given a non-final rejection in Nov, 2023 (<https://patentcenter.uspto.gov/applications/17626700/ifw/docs?application=>). While we are not qualified to comment on the legal aspects of the differences between our method and the currently rejected patent, in terms of engineering and scientific differences, our method differs in significant ways from the Mocus system as described both in the patent and in the preprint.

Mocus is a stereoscopic VR simulator composed of separate display “wings” that are individually positioned for each eye and secured to the animal head-clamping mechanism, distinguishing itself from prior VR designs that are single-piece headsets and that can be attached to freely-walking animals. MouseGoggles on the other hand is a single piece design similar to prior VR/AR headsets and that is not attached to the head-clamp. MouseGoggles is also built on entirely different hardware and software platforms than Mocus (MouseGoggles

with a Raspberry Pi computer, Godot game engine, and SPI-based displays, compared to Moccus with a Windows PC, Unity engine, and HDMI-based displays); in fact, there is not a single piece of hardware or software in common between the two systems. We further successfully demonstrate in situ tracking of eye gaze direction and pupil diameter, a feature the neuroscience community has explicitly requested from us. In this manuscript, our focus is reporting on the design and results from our mouse VR system, and we must leave questions about the impact of this patent to the US Patent and Trademark Office, and comparable entities in other countries.

Comment 5: The authors should improve the online documentation that is cited in the manuscript in order to better facilitate replication of this open source method.

Response: We thank the reviewer for such detailed and helpful notes regarding the online documentation. We have updated the repository [REDACTED] with all details requested and listed in the following comments.

Comment 6: Please upload not only the STL files but also the STP files or IPT drawing of the CAD designs for easier modification

Response: For all custom 3D designed parts, both .stl files and .step files have been uploaded to facilitate modification.

Comment 7: The bill of materials mentions all parts in [cm] however Figure 2 shows them in [mm] – please update the bill of materials and use only one unit of measurement to avoid confusion

Response: The bill of materials has been updated to show 3D print sizes in mm to stay consistent with the rest of the repository and manuscript.

Comment 8: Recommendation for a future revision: Reconsider the material that was used in the printing. If the parts will be adjustable or moveable at some point, resin material will become brittle quickly.

Response: We thank the reviewer for this recommendation. In our online assembly instructions, we have added new recommendations for different materials, 3D printers, and print settings to use. To successfully print the small features of our eyepiece enclosure, we recommend high resolution printers capable of <0.2 mm layer resolution. Low-cost SLA printers (e.g. Photon Mono X) using standard UV-cured resin can be very cost-effective and successfully print all parts used by MouseGoggles. Parts produced this way have remained intact after over 2 years of use in our hands. For more rigid and longer-lasting parts, a high resolution FDM printer (e.g. Ultimaker) can be used with stronger and tougher thermoplastics such as PLA, PETG, or ABS.

Comment 9: Upload not only gerber files but also the schematic and PCB file for easier modification

Response: For all custom PCBs, both the schematic and layout files as well as the Autodesk Eagle project file have been added to the online repository to facilitate modification.

Comment 10: Include the fabrication specifications for the PCBs: board thickness, copper thickness and surface finish

Response: Fabrication specifications for custom PCBs have been added to the online repository, including the number of layers (2), board material (FR-4), board thickness (1.6 mm), copper thickness (0.0348 mm), and surface finish (HASL with lead).

Comment 11: In MinibrdV1, the copper to copper spacing is only 0.06 mm – for a future revision I recommend updating the layout to increase the minimum copper to copper spacing to at least 0.2 mm; this simplifies demands on production.

Response: In MinibrdV1 (now renamed to MouseGoggles MiniBrdAA 1.1 as part of a comprehensive version naming structure), the minimum copper to copper spacing has been adjusted to 0.2 mm. All other PCB layout files have been verified to fit this requirement as well. Thank you for this valuable suggestion.

Comment 12: In MinihatV1, there are 90 degree traces and these will form acid traps that cause corrosion – for a future revision I recommend updating the layout with traces only at 45 degrees

Response: In MinihatV1 (now renamed to MouseGoggles MiniHatAA 1.1), all 90-degree traces have been removed and replaced with 45-degree traces. Other layout files have been checked for these as well. Thank you again for helping us with these PCB design subtleties.

Comment 13: The driver for sending the images via SPI to two small screens is from an external project (fbcp-ili9341 and fbcp-st7789) which seems to be well documented, no action required.

Response: We thank the reviewer for this comment, and again for thoroughly checking the open documentation. Our selection of this driver was in keeping with the open-source, well-documented ethos of this project.

Comment 14: Drivers for detecting the movement of the ball.

The movement is transmitted to the PC/RasPi as if it were a PC mouse. The ADNS3080 sensor is used. "mouseVRheadset_controller_V4.ino" is not commented in detail. This is so small that this is not of great importance. Action: add a few explanatory sentences to facilitate faster orientation.

Response: We thank the reviewer for pointing out our lack of documentation on the treadmill system. Since our focus has been on the development of visual display hardware and software, we mistakenly omitted some documentation on the treadmill system software we've developed for MouseGoggles compatibility. We have now added new documentation describing the software for two treadmill systems: the spherical treadmill system originally from Harvey et al, Nature 2009 (<https://www.nature.com/articles/nature08499>) that was described in our original manuscript, as well as a linear treadmill from Arnold, JRC, 2023 (<https://www.janelia.org/open-science/low-friction-rodent-driven-belt-treadmil>) which we have recently implemented with MouseGoggles, and is included in our revision (Fig. 4). Our new documentation describes where the treadmill hardware is described and/or can be purchased, and details how to update the treadmill's microcontroller firmware for compatibility with the MouseGoggles system.

Comment 15: The purpose and use of the files in the "ADNS3080 debugging" folder must be explained. While these are only for debugging, how and what they are used for is not clear. This is also because these data have no comments. Action required: Explain and comment.

Response: A description of this debugging folder has been added to the spherical treadmill documentation described above, and additional in-code comments have been added to the scripts within that folder.

Comment 16: The experiments with 3D glasses use the Godot engine. Some examples are provided. But you need some knowledge about this engine, especially because the experiments run completely within the engine. This engine is not only responsible for the representation of the 3D world, but also for the process and the logging of the data. The examples make it easier

to get started, but still require some familiarisation with Godot. Recommended action: the authors might think about how they could further facilitate getting familiar with this part of the code.

Response: Since the Godot game engine has not been previously used in neuroscience applications (unlike Unity), adding a tutorial or familiarization document for the Game engine is an excellent suggestion. To the online repository, we have added a new document to function as both a general 'Introduction to Godot' as well as an overview of the specific Godot files created and used for MouseGoggles experiments. This includes walking the reader through the Godot game engine editor and scripting environment, organization of different types of game files, highlighting some commonly used features and settings, and walking through a typical experiment script and logged data file. We also include seven example VR environments that can help get users started.

Comment 17: The glasses are also available in a variant for "Monocular Display". This uses the Teensy board to control a small display. "GC9307_teeny_GFX.ino" has a lot of comments on single lines, but a more general documentation to understand the context of the parameters would be nice. Especially because there are so many parameters. Action required: provide a better general documentation and include a table of all parameters and their use.

Response: Since the Monocular Display (now referred to as MouseGoggles Mono) is built on a different hardware and software platform as the binocular MouseGoggles Duo (Arduino graphics library + Teensy microcontroller, vs Godot game engine + Raspberry Pi computer), we agree that it makes sense to have comprehensive documentation for the MouseGoggles Mono code and control system. We have added new documentation which describes a general overview of the system, the different operational modes of the system, and describes and lists the commands and parameters which can be sent to the microcontroller to display patterns on the monocular display.

Comment 18: While the software is neither very large nor a complex project, with its current level of documentation it still poses a challenge to find ones way around. This does not apply to the external SPI monitor driver. This part alone is much larger and more complex than MouseGoggles.

Response: With the extensive documentation added and described above, including both specific technical details as well as general overviews of the software and systems, we believe that the online repository is now easier to navigate and understand, and users will be better able to replicate and modify this tool for their own purposes.

Comment 19: 551 check sentence for completeness: ... last bin excluded due the mouse's constrained position ...

Response: We have rewritten this sentence for completeness: "*All licking data was binned by location into 5-cm wide bins, with the first and last bin were excluded due to the mouse's constrained position away from the walls*". (line 739-741).

Comment 20: 555 For ease of understanding change: "All licks not occurring following a reward delivery were defined as "exploratory licks". TO Licks at other times than after a reward delivery

Response: This suggested change has been made (line 744-745): "*All licks occurring at other times than after a reward delivery were defined as "exploratory licks"*".

Comment 21: 619 a 2nd cohort of mice were tested -> was tested

Response: This typo has been fixed (line 808).

Related to the typographical errors mentioned above, we have made additional corrections to our methods section after identifying missing or confusing information:

- Missing information about post-operative analgesia after the surgical procedure for head-fixed behavior has been added (lines 530-531).
- Missing information about the concentration of AAV-GCaMP6s bolus injection for V1 calcium imaging has been added (lines 538).
- The section on habituation for head-fixed behavior has been rewritten for clarity, with added details for habituation on our newly included linear treadmill (lines 660-671).

Comments from Reviewer 3:

Comment 1: Virtual reality has become a powerful tool for studying a variety of brain functions in head-fixed rodents. Current implementations of rodent VR involve large displays that not only create difficulties for the experimenter, but also do not provide an immersive experience for the animal, largely due to their expansive visual field. Isaacson, Chang, et al. designed a head-set based VR system to overcome current constraints. They provide evidence that their system solves many of the practical issues with VR, including reduced light pollution for imaging methods and reduced cost. They use neural recordings to show that the visual tuning properties of neurons are comparable to results obtained with standard techniques, supporting the quality of image presentation. Finally, they provide evidence that the VR experience for the mouse is immersive through effective behavioral training in reward and innate fear behaviors.

This work represents a very useful advance for both the quality of rodent VR experience and the ability of individual laboratories to implement VR systems, especially given the low-cost and open source design. Overall, the authors provide a straightforward and convincing case for the improvements this system makes over traditional rodent VR systems.

Response: We thank the reviewer for these positive comments.

Comment 2: The manuscript could be substantially improved by addressing the following points: The system provides 230 deg of field coverage in the horizontal azimuth, with 140 deg per eye and 25 deg of binocular overlap. The authors should report measured values of the mouse visual field from previous studies in the main text, and discuss the limitations of the system in obtaining full coverage in the Discussion. Also, the approximate elevation covered by the system should be reported in the main text.

Response: We agree with this suggested change and have moved information on mouse visual field estimates and our coverage of that field from the methods to the main text. We also report approximate elevation covered by the system (140 deg) and the headset configuration that was predominantly used in this manuscript (15 deg pitch, for -55 to +85 deg maximum stimulated elevation relative to the horizon). These details have been added on lines 55-58. Details on different pitch positions of the headset and the estimated visual field coverage of each position has been added in Supplementary Fig. 7. Relating to this, we noticed an error in our plotting of the estimated mouse visual field, which is now correctly plotted as a 180x140 deg ellipse (Fig. 1e, Supplementary Fig. 6d, and Supplementary Fig. 7c).

We have also added discussion on the strengths and limitations of our relatively simple optical design (lines 159-162). The main limitation being that FOV coverage is limited to 140 deg, whereas the main strengths are a simple, inexpensive build requiring no custom lenses, and a system which is robust to eye position variations and does not require precise eye

alignment protocols. We have added a new supplementary figure demonstrating the system's performance by imaging gratings through the back of an enucleated mouse eye, showing similar performance to a traditional flat monitor (Supplementary Fig. 1).

Comment 3: How does the 130 ms input to display latency compare to other rodent VR systems? My understanding is that latencies greater than 20ms are detectable to humans, so it seems possible this relatively long latency may limit the immersive experience for the mouse. The authors should discuss this.

Response: We believe that total system latency is an important consideration for any closed-loop system. To our knowledge, most existing rodent VR systems do not report end-to-end latency, thus it can be difficult to compare. Among modern, high-end human VR headset systems, mean total latency has been reported to range from 21-42 ms (<https://pubmed.ncbi.nlm.nih.gov/36217006/>), where the longest latencies were found during periods of rapid movements. Among VR systems used in animal model neuroscience (specifically, those that report total latency), total latency ranges more broadly, with values as small as 2 ms under restrictive conditions (Isaacson et al, BioRxiv 2022: <https://www.biorxiv.org/content/10.1101/2022.08.02.502550v1>) to ~100 ms for general-purpose VR (Madhav et al, J Neurosci Methods 2022: <https://www.ncbi.nlm.nih.gov/pmc/articles/PMC9178503/>). Our panoramic projector VR system, after optimization, measured at ~90 ms latency. As for what latencies are needed or desired for head-fixed animals to be "immersed" in the VR world, we aren't able to say, as to our knowledge very little has been published on the effects of latency on neuroscience VR applications. One might speculate that long latencies are workable for closed-loop feedback from slow behaviors, such as mice walking on a treadmill, but could be problematic with fast behaviors such as eye movements.

Without optimization, we measured MouseGoggles latency at ~130 ms. However, there are opportunities to substantially reduce this latency in the display driver. Using "adaptive display stream updates", where only pixels which changed from the previous frame are updated, total latency can be drastically reduced. This feature is supported by the display driver used by MouseGoggles, and has been demonstrated to reduce total latency on a single display comparable to that used by MouseGoggles to under 20 ms (<https://github.com/jui/fbcp-iii9341#about-input-latency>). This feature has not yet been implemented for dual displays, though this is an area we are actively working on, and we hope to have this feature implemented and latency reduced in the near future (we will update our online repository accordingly). We have added these details on total latency and potential future optimization in the main text (lines 156-159) and the methods (lines 417-420).

Comment 4: The measurements in 1b-d are in the range of 0-70 deg, but the authors report monocular coverage of 130 deg. Is there some reason why they did not model the full coverage, which is nearly twice the range in the figure?

Response: We thank the reviewer for pointing out this confusion. The 130 deg value comes from the total horizontal extent of the visual field coverage, which includes both left and right directions from straight ahead (70 deg in one direction and 60 deg in the other due to the "chipped edge" of the 240x210 circular display). In Fig. 1b-d, we modeled the viewing angle in a single direction up to 70 deg; since the display and lens configuration is radially symmetric, this modeling will be identical in all other directions. We have reworded the text of the figure legend (lines 196-197), referring to the viewing angle as one side of the total maximum field of view coverage (140 deg) to reduce this confusion.

Comment 5: The modeling of the light reaching the eye from the screen/lens is very useful. It would also be useful to have a measurement of the performance of the actual system, e.g., by projecting onto a model or real mouse eye.

Response: As suggested, we have now imaged the display's projection on the back of enucleated mouse eyes during the presentation of drifting gratings, comparing the image generated by a MouseGoggles eyepiece to that of a traditional flat monitor positioned further away (10 cm). We found that the image produced onto the plane of the retina by MouseGoggles is qualitatively comparable to that of the monitor, except that MouseGoggles is able to cover a much larger area of the eye's FOV. Furthermore, since one theoretical benefit of our relatively simple optical design is that the eye is not required to be precisely positioned relative to the lens (with the display positioned at the lens' focal distance, rays extending from the lens to the eye should be close to parallel), we measured the robustness of the system to minor changes in eye position by shifting the eye relative to the eyepiece and imaging the projection. We found that changes in position of up to ~2 mm orthogonal to the optical axis of the lens and produced no clear distortion of the image projection on the back of the eye adding experimental support to the theoretical robustness of our design. These results are shown in Supplementary Fig. 1 and referenced in the main text (lines 50-52) and methods (lines 425-439).

Comment 6: The comparison of light pollution between the monitor and monocular display is quite striking. It would be ideal to see a similar direct comparison of the two systems for the visual tuning properties in Fig2d-i, especially since the stimulus used for receptive field mapping is different from the referenced study (Niell & Stryker, 2008).

Response: Related to this comment, we describe new experiments and analyses on light pollution comparing MouseGoggles to a traditional monitor in response to Reviewer 1, Comment 6, above. Regarding differences in stimuli used to map visual responses, the stimuli we used to measure visual tuning properties (Fig. 2d-i) were designed to closely match prior studies so that they could be considered close to direct comparisons. We do agree with the reviewer that our receptive field mapping stimuli is somewhat different than that of the referenced study, though we believe this is not a significant difference and our original description of our stimuli could give the impression that the stimuli were more different than they actually were. The study we designed this receptive field mapping test on (Niel & Stryker, 2008) used a 4x8 deg bright bar sweeping across the visual field, repeated in 4 directions and at 8 different azimuths and elevations. Our study used a 3.8x7.6 deg bright bar (the closest we could get with our display's resolution), repeated across 5 different azimuths and elevations to reduce the total experiment time required – since our experiments began with a search to locate receptive field centers, a 5x5 grid was sufficient in our experiments to capture many neurons' receptive fields. There were minor differences in the presentation order of the stimuli across the visual field: Niel and Styker presented a continuous sweep in one direction across 8 positions before switching to a new direction and position, while we presented sweeps in 4 directions at a single position before switching to a new position. The amount of the visual field stimulated in each data bin was nearly identical to the study from Niel and Styker, which is expected to be the main determining factor in the receptive field size measurement. A likely source of confusion in our original description of the stimuli was that we described our stimuli as gratings rather than bar sweeps. However, each of these "gratings" had a 7.6 deg spatial wavelength and was presented at 1 Hz for 0.5 s, which is essentially just a bright bar sweep across the 7.6 deg position, similar to the study from Neil and Stryker. We apologize for this confusing description and have now adjusted the text to describe this stimulus as a bar sweep (lines 85, 587-589), but if there are still any concerns about this comparison, we are happy to revisit or repeat this experiment with a new protocol.

Comment 7: How do the tuning properties of CA1 place cells compare to other mouse VR studies and freely moving mouse studies (e.g. place field width)?

Response: Since the properties of place fields in rodent dorsal CA1 are known to be modulated by the size of the virtual environment (e.g. Harland et al, Curr Bio 2021:

<https://www.sciencedirect.com/science/article/pii/S0960982221003420#bib8>) and richness of the local visual cues (Tanni et al, Curr Bio 2022:

<https://www.sciencedirect.com/science/article/pii/S0960982222010089#bib16>), and since our virtual linear track was visually distinct from those in prior studies, a direct comparison may be difficult to interpret. However, we agree that CA1 place cell tuning properties are useful for interpreting how the VR experiment was encoded by place cells, so we have added new quantifications to our manuscript: proportion of neurons that exhibited place-like characteristics for each session, place field width in the virtual space, and information rate for each place cell. These quantifications have been added to Fig. 2 (panel n), and references to these properties have been added to the main text (lines 99-102). Details on how these new properties were quantified have been added to the methods (lines 702-707).

Comment 8: In comparing the behavioral responses to looming stimuli on MouseGoggles vs. a traditional projector system, was the projector system oriented above the mouse? Based on the methods, it appears the projector reached a maximum of only 64 deg elevation, so is this the presumed reason that a response was not elicited? The ability to adjust elevation with MouseGoggles is indeed useful, but ideally the behavior would be compared to a projector system with similar visual field coverage. If it is experimentally infeasible to adjust the projector system to cover a comparable region of the visual field, the authors should state this. Otherwise, the experiment should be conducted with comparable overhead coverage between the two systems.

Response: The projector-based VR system we used featured a custom conical screen that extended significantly above the horizon, with only a circular cutout above the mouse to fit a microscope objective for 2-photon imaging. Even though the projector had a large vertical extent, the reviewer is correct that the FOV coverage of MouseGoggles extends even further. This increased overhead coverage was likely not meaningful for the looming experiment since the looming object approached from a 45 deg elevation, which is easily covered by the 64 deg elevation of the projector screen. However, to be sure, during the newer looming experiments with eye and pupil tracking (shown in Fig. 4 and Supplementary Video 3) we blacked out the top section of the screen (above 64 deg elevation) and we observed similar startle responses from these mice. This detail on the restricted elevation has been added to the methods section (lines 819-821).

Comment 9: It appears that video of the mouse was acquired during the looming stimulus presentation (Fig3f and methods). The methods clearly state the responses were manually scored, but the main text and figure do not mention this, and the line fit to the mouse's back in Fig3f might lead the reader to assume the quantification of behavior was achieved with computational video analysis. Is the resolution of the video sufficient to perform the startle analysis? If so, the authors should quantify the behavior using image analysis methods or markerless pose estimation. If not, the authors should clearly state that in the text/ Fig3g that the quantification was performed by two human observers. The authors could also plot the velocity of the animal measured via the spherical treadmill aligned to the onset of the looming stimulus.

Response: We thank the reviewer for pointing out this omission in our text. One complication we observed during early attempts at automatic quantification and detection of looming reactions is the significant variability in mouse behavior prior to the looming stimulus; head-fixed mice may be stopped, walking, running, or grooming immediately prior to the loom. Therefore, we determined that identifying reactions to looming stimuli are most clearly assessed through

manual behavior scoring. We have clarified this approach in the main text and Fig. 3 legend (lines 121, 249-252).

In inspecting and plotting the spherical treadmill velocity aligned to the looming stimulus, we found it impossible to detect startle responses in terms of treadmill movements, seemingly due to the near-constant movement of the low-friction treadmill and the relatively small startle-movement transferred to the ball. However, in the new looming experiments with eye and pupil tracking which were performed on a more stable linear treadmill, plotting the linear treadmill velocity showed a clearer response from the looming stimuli (mice back up a bit!). This new data is shown in Fig. 4, referenced in the main text and described in the methods section (lines 137-142, 815-839).

Comment 10: Does this system permit the measurement of eye movements? Given that mice move their eyes (though somewhat infrequently) while head-fixed, it would be ideal to measure these movements during stimulus presentation. The authors mention that pupil tracking would be feasible with this system; how would this be achieved?

Response: Although it was a lot of work, we thank the reviewer for challenging us to achieve this goal. The ability to measure eye and pupil dynamics during head-fixed VR has been the most requested feature for future versions of the MouseGoggles system, and we are excited to add this capability in our revised manuscript. As we show in new figure panels (Fig. 4), the MouseGoggles system does indeed permit eye and pupil measurement during VR presentation through the use of fully-integrated infrared imaging inside the eyepiece. Due to the simple optical design of MouseGoggles, enough space exists within the eyepiece to fit an angled hot mirror, which allows visible display light to pass from the VR display to the mouse eye while IR illumination can reflect to the eye and back toward a mini-IR camera. This new design, which **still** only uses off-the-shelf parts, enables clear imaging of the mouse eye and pupil during visual stimulus presentation. This optical strategy for simultaneous eye/pupil monitoring and VR presentation might not be possible with the other recent mouse VR headsets, Mocus and iMRSIV, due to their more complex optical design and reduced space within the eyepiece enclosure to fit additional lenses or mirrors. The iMRSIV publication specifically identifies a lack of eyepiece space as a hindrance to simultaneous eye monitoring in their system, and notes that eye tracking would be highly beneficial for future VR headsets.

To demonstrate this new feature, we performed infrared eye imaging of head-fixed mice walking on a linear treadmill during the looming visual stimulus experiment. Position of points along the eye and pupil were tracked using DeepLabcut to enable measurement of both eye orientation and pupil diameter. From this experiment, we observed clear eye tracking towards the overhead looming objects and noted a pupil dilation response following the looming stimulus that diminished with additional repetitions of the stimulus, similar to the habituating startle responses. These new results are shown in Fig. 4 and discussed in the main text and detailed in the methods section (lines 127-142, 441-489, 815-839).

Comment 11: Face/whisker movements contribute significantly to neural activity. Does this system allow for these measurements, or do the MouseGoggles fully obstruct the view?

Response: Face movements can indeed be seen with the MouseGoggles system if a camera is mounted in front of the mouse and below the headset (e.g. supplementary Video 1), where views of the nose, mouth, and many lower whiskers are unobstructed. However, upper whisker movements are significantly impaired as many upper whiskers during whisking behavior cross in front of and near the eyes. These details have been added to the main text (lines 154-156).

Comment 12: If possible, the experimenters should quantify whisker contact with the system. Do any of the whiskers touch the system, and if so with what frequency? Extensive whisker

contact with the system would presumably degrade any immersive experience the mouse may have, and would make it difficult to perform some of the multisensory experiments the authors propose in the Discussion.

Response: We thank the reviewer for this excellent suggestion, we agree that quantifying whisker contact is important to characterize our system and its limitations. We have added a new supplementary figure quantifying the whisker contact depending on different orientations of the headset. Depending on the headset pitch, slightly more or less of the head-fixed mouse's whiskers will be occluded by the headset. In a near-flat position, where the headset is pitched 15 degrees upward from the horizon (a typical orientation used in this study), approximately 60% of whiskers are blocked by the headset. With significantly increased positive headset pitch (e.g. 45 deg), where more of the visual field above the horizon is covered, some frontal whiskers become unblocked. These considerations should assist future users in selecting a headset position that will enable their desired experiments. These new results are shown in Supplementary Fig. 7 and referenced in the main text (lines 154-156).

Comment 13: How do the authors calibrate the system with each individual mouse, given e.g., the inherent variability in headplate placement?

Response: Due to the simple optical design of the MouseGoggles headset, the system is robust to minor offsets of the mouse eye from the center of the lens. To demonstrate this feature more clearly, we have added Supplementary Fig. 1 showing the performance of the optical design by viewing the display's projection on the back of an enucleated mouse eye. By adjusting the position of the eye relative to the Fresnel lens surface center, we found that small lateral eye displacements (<2 mm) caused little noticeable distortion of the display image. In our experience, manual adjustment of the optical posts and angled post clamps until both eyes were roughly centered inside each eyepiece routinely achieves binocular eye positioning well within this range. This may be a unique benefit of the MouseGoggles system compared to the Mocus and iMRSIV systems which use more complex lens arrays and are described as being sensitive to eye position, and whose methods describe a more precise eye alignment protocol. We have added a discussion on this issue to the main text and to the methods section (lines 159-162).

In our surgical preparations the headplate placement did not significantly vary in tilt, therefore the only adjustment necessary for accurate positioning of mice relative to the headset was manually translating the headset up to the mouse's eyes, with no sideways tilt required to roughly center both eyes inside the eyepieces. We have added this detail on headset positioning to the methods section (lines 666-670). In the case that a surgical preparation results in a significantly tilted headplate (either in pitch or roll directions), a further calibration might be needed in the VR software if it is desired to match the virtual "eyes" to the headset orientation and position. While we did not have to make this kind of adjustment in our experiments, we have updated the VR experiment scripts in the online repository XXXXXXXXXX so that both headset pitch and roll tilt angles can be easily matched by the game engine.

Comment 14: Given that monocular cues should be sufficient for most experiments performed in the manuscript (formation of place cells, responses to looming stimuli, reinforcement learning), an "immersive" experience may not be requisite for the results obtained here. A more convincing test that the animal is having an immersive experience would be ideal, e.g., behavior with a virtual cliff showing that animals stop to avoid falling over the edge.

Response: Due to the lack of startle responses to looming visual stimuli from head-fixed mice in our panoramic projector-based VR system, we believe that even wide-field binocular visual cues are insufficient to cause startle responses, and monocular cues have not been shown, to our knowledge, to generate a startle response. Thus, an immersive virtual experience seems to be necessary and is successfully demonstrated by our MouseGoggles system. However, we

agree with the reviewer that a visual cliff experiment may be an excellent test that only a binocular, immersive, and stereoscopic VR system could replicate. As we detailed more fully in response to reviewer 1, comment 4, the current understanding of the salience of stereoscopic vision in mice is limited, making this more a topic for exploration with stereoscopic VR systems than a robust testbed for establishing the degree of immersion. The only experiment we are aware of that demonstrates mouse's innate use of stereo cues is in the angled pole descent test (Boone et al, Curr Bio 2021: [https://www.cell.com/current-biology/fulltext/S0960-9822\(21\)00272-4](https://www.cell.com/current-biology/fulltext/S0960-9822(21)00272-4)). This test is required over a typical flat visual cliff experiment as the downward angle ensures that the cliff appears well within the mouse's binocular overlap region, where stereo vision has been shown to apply. MouseGoggles, combined with a custom pitching treadmill, could provide an excellent toolkit for testing the salience of stereoscopic vision in mice, but is beyond the scope of this manuscript. Our additions, in this revision, of implementation with a compact linear treadmill (Fig. 4) and with closed-loop feedback from headset rotation (Supplementary Video 4), should facilitate the development of this kind of VR experiment in the future.

Dear Dr. Vogt,

We thank you and the reviewers for providing feedback on our revised manuscript, "MouseGoggles: an immersive virtual reality headset for mouse neuroscience and behavior". We appreciate all the positive feedback on our recent changes and the final comments to ready the manuscript for publication. Below is our point-by-point response to the remaining reviewer comments.

Comments from Reviewer 1:

Comment 1: Authors addressed all my comments, I would fully support publishing this paper.

Thank you for your positive feedback and support for the publication of our paper, and for the excellent suggestions in the previous comments.

Comments from Reviewer 2:

Comment 1: The authors have made quite some effort to respond to my concerns. [referencing previous comment 14] The "Hardware" directory has been renamed to "Other Hardware". This was a good decision, as it helps to differentiate between MouseGoggles and other hardware mentioned in paper. The added documentation should now enable and simplify the initial setup of treadmill and the sensors.

Thank you for your positive feedback.

Comment 2: [referencing previous comment 15] Still no comments in the code. But now it is explained how these files should be used and their purpose. Since the files are small, this seems fine now.

In addition to the explanation of the purpose of the debugging files in our online repository, we have now added comments to the "ADNS3080 debugging" code for each file individually, as requested earlier. Sorry for the omission, we mistakenly had not pushed an update after our last response, but it has now been updated and verified.

Comment 3: [referencing previous comment 16] The new short "Introduction to Godot" should indeed make it a little easier to get started. Even if there is still a relatively large hurdle here, as Godot has to be learned. But the experiments seem to be both very small and very similar, so that customisation should be possible.

Thank you for your feedback on the "Introduction to Godot" file. We understand that there is still a learning curve, as there will be with any experiment design program or game engine, but we hope that our aim to keep the example 3D experiments simple will also make them [relatively] easy to understand and customize.

Comment 4: Small note and suggestion for a further small change: it should be noted which version of Godot is used here. I opened the project with 4.2.2 and had some error messages in the script editor. This happend although the documentation explicitly calls for the last version of Godot. However, this probably refers to the latest 3.x version.

The version of game engine software used by in this manuscript (Godot 3.2) has now been added to the "Introduction to godot" document in our online repository.

Comment 5: [referencing previous comment 17] The description of the hardware and the hardware assembly instructions should now be sufficient to build the system. And the software description/documentation should help to setup the system. [referencing previous comment 18] I agree.

Thank you for your agreement and positive feedback on the hardware and software documentation, and for all your help in improving the online resource.

Comments from Reviewer 3:

Comment 1: The updated manuscript provided by Isaacson, Chang et al. entitled “MouseGoggles: an immersive virtual reality headset for mouse neuroscience and behavior” represents a substantial improvement over the initial manuscript submission. The authors responded to the reviewer comments with a significant amount of work that addresses the issues with the original manuscript, including:

- **Relationship of the system to the total visual field**
- **Discussion of latency and steps toward reducing it**
- **Measured the light reaching the eye**
- **Clarification on visual stimuli**
- **Quantification of place cell properties**
- **Improved behavioral measures during looming behavior**
- **Characterization of whisker obstruction**

The heroic addition of pupillometry also sets this system apart from other emerging immersive mouse VR systems, making this system particularly useful to neuroscientists who also want to track pupil position and size.

We are grateful for your thorough and positive evaluation of our manuscript. We appreciate the acknowledgment of our revisions and additions, including the incorporation of pupillometry which was a particular challenge but one we also believe will be highly useful for behavioral neuroscientists. Thank you for the earlier feedback which motivated these improvements.

Comment 2: I have only a minor comment, which is that the authors should consider mentioning a finding from Meyer et al., Curr Bio 2020, that might explain the eye movements in response to the looming stimulus. Eye movements in headfixed mice appear to result primarily from attempted head movements (see Fig S5 in that paper), so one interpretation of the results here is that the animals are attempting to escape by moving their heads, which results in a saccade. While a subtle point, the authors’ interpretation that the animal’s gaze is being directed toward the stimulus is more likely due to an artifact of the animal’s inability to move its head freely.

Thank you for this insightful suggestion. Since head-fixed eye movements have been shown to be related to attempted head movements, we have added this suggested interpretation of eye movements during our virtual looming experiment and added a reference to Meyer et al., Curr Bio 2020 (lines 152-154).

We appreciate the valuable feedback from all reviewers and hope that these revisions address all remaining concerns.

Sincerely,
Matthew Isaacson
On behalf of all authors